# The perpetual fragility of creeping hillslopes

Nakul S. Deshpande[1], David J. Furbish[2,3], Paulo E. Arratia[1,2] & Douglas J. Jerolmack [1,4✉]

Soil creeps imperceptibly but relentlessly downhill, shaping landscapes and the human and ecological communities that live within them. What causes this granular material to 'flow' at angles well below repose? The unchallenged dogma is churning of soil by (bio)physical disturbances. Here we experimentally render slow creep dynamics down to micron scale, in a laboratory hillslope where disturbances can be tuned. Surprisingly, we find that even an undisturbed sandpile creeps indefinitely, with rates and styles comparable to natural hill-slopes. Creep progressively slows as the initially fragile pile relaxes into a lower energy state. This slowing can be enhanced or reversed with different imposed disturbances. Our observations suggest a new model for soil as a creeping glass, wherein environmental disturbances maintain soil in a perpetually fragile state.

[1] Department of Earth and Environmental Science, University of Pennsylvania, Philadelphia, PA, USA. [2] Department of Earth and Environmental Sciences, Vanderbilt University, Nashville, TN, USA. [3] Civil and Environmental Engineering, Vanderbilt University, Nashville, TN, USA. [4] Department of Mechanical Engineering & Applied Mechanics, University of Pennsylvania, Philadelphia, PA, USA. ✉email: sediment@sas.upenn.edu

The shapes of hills encode a signature of tectonics, climate, and life, through the influence of these processes on sediment transport[1–4]. Soil fails by landslides on the steepest slopes, leaving telltale scars on the landscape. Below the angle of repose; however, soil-mantled hillslopes are characteristically smooth and convex[3,4] (Fig. 1). Although this soil is considered a solid, it appears to flow over geologic time in a process called soil creep[5,6]. What is the mechanism for granular motion below the angle of repose? This has been speculated on for over 100 years[7,8]. Hillslopes are perpetually disturbed: bombarded by seismic waves, thermal cycles, wetting and drying, and bioturbation[9–14]. Current hillslope creep models trace their origin to Culling[15], who envisioned that the net effect of these disturbances was to inject porosity, which facilitates particle motion. He also recognized that porosity, and the associated particle activity, must diminish with depth. In the continuum limit, Culling proposed a diffusion-like relation between sediment flux and topographic gradient, that has been elaborated on by many authors and implemented in virtually all landscape evolution models[3,4,6,14,16,17]. The hypothesized grain motions in Culling's model, however, are inconsistent with known granular mechanics (Supplementary Note 1); moreover, these motions have never been experimentally examined. Researchers have begun to recognize the need to understand grain-scale dynamics, in order to derive physically informed models of soil mixing and transport on hillslopes[18,19]. Decades of tracer measurements have produced coarse profiles of soil displacement on hillslopes[5] that are generally exponential[5,20,21] (Fig. 1); the slow and erratic nature of creep, however, has prevented direct observation of grain motions in the field. The canonical hillslope laboratory experiment of Roering and colleagues[22] showed how acoustic noise can induce grain motion below the angle of repose; however, our reanalysis indicates that grains were actually fluidized, rather than subcritically creeping (Supplementary Note 2).

Creep has long been recognized in the context of dense granular flows, which transition at depth to a slow creep regime characterized by intermittent and apparently random particle motions[23,24] and exponential velocity profiles[25,26]. A flowing layer at the surface is not necessary, however, to induce creep[27,28]. Observations in a progressively tilted sandbox showed that, on approach to the angle of repose, sporadic and localized grain motions became more frequent and eventually linked up to affect yield[27]. Granular simulations have reproduced these behaviors without any imposed disturbances[24], and shown that low-amplitude noise does not qualitatively change the picture[24,29]. There is emerging evidence that granular creep shares deep similarities with other amorphous solids[24,30,31], such as glass, where creep is associated with sub-yield plastic deformation in response to an applied stress[32]. A unifying characteristic of amorphous solids is that they are fragile: any particle configuration is metastable, and very small perturbations can lead to structural rearrangements[30,31]. The origins of this metastability are locally weak zones, which yield in response to the globally applied stress. These creep motions are manifested as spatially heterogeneous, mesoscopic (length $\gg$ grain size, $d$) zones of strain[32]. In glasses, relaxation by plastic rearrangements leads to aging; rigidity increases with time, leading to a slowdown in creep rates. This decline in plasticity can be reversed by rejuvenation,

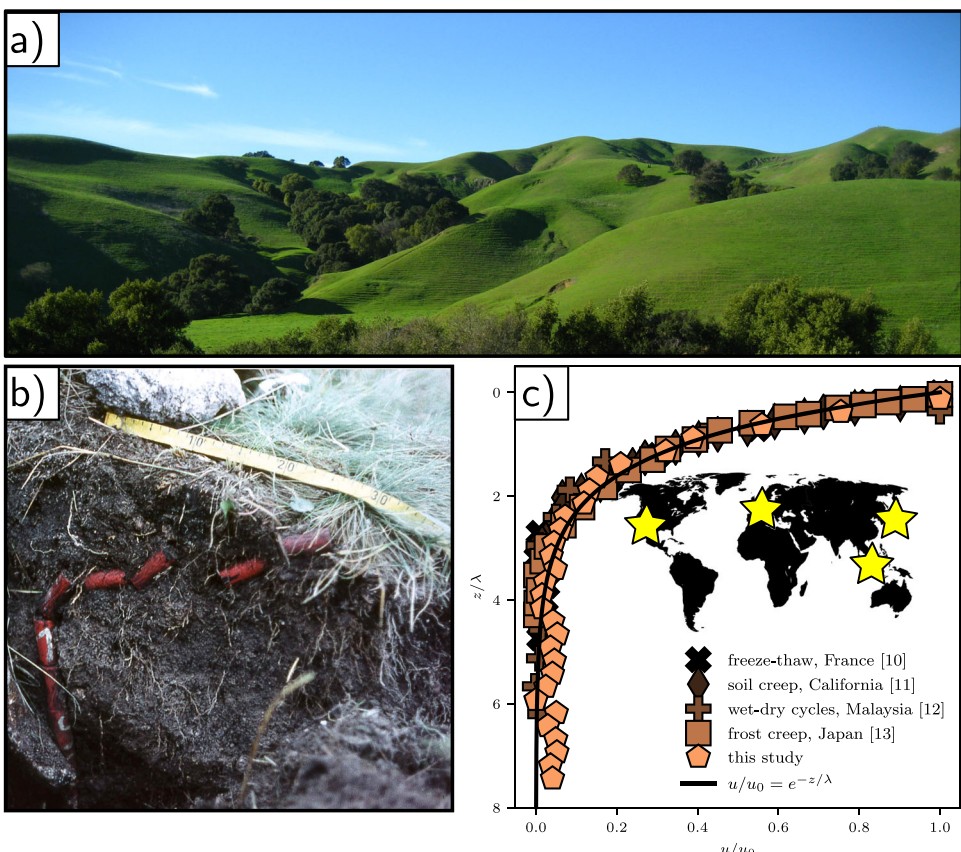

**Fig. 1 Soil creep observations and scales. a** Canonical soil-mantled hillslopes, Briones Regional Park, California. **b** Excavated Young Pit indicating the displacement of tracer pegs over a 17-year interval in the Sudetes Mountains, Poland (image: Alfred Jahn). **c** Compilation of soil deformation data from four studies and field environments, originally compiled by Roering[21]: freeze–thaw cycles near Strasbourg, France[10]; wet–dry cycles in Stanford, California[11]; wet–dry cycles in Kuala Lumpur, Malaysia[12], and freeze–thaw cycles in the Japanese Alps[13]. Data are collapsed by a normalized exponential function (Supplementary Notes 5 and 6). Also shown is a measured strain-rate profile from our experiments; see text for details.

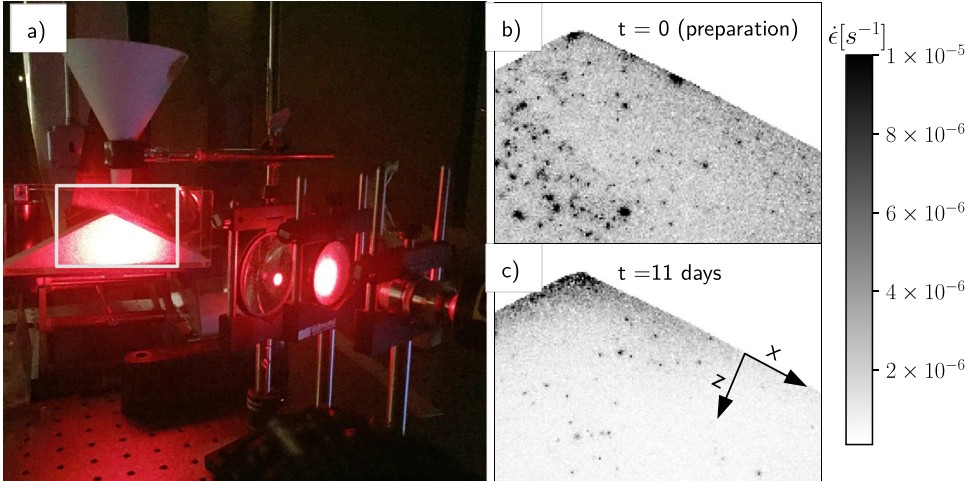

**Fig. 2 Experimental setup and phenomenology. a** DWS setup: an expanded laser beam is projected onto a granular heap; a camera collects speckle images from within the white rectangle. **b** Strain-rate map at $t = 0$, the moment after the pile is prepared, and (**c**) 11 days after preparation showing that particle activity persists.

typically by changing temperature or applied external forces[33]. Theorists have proposed that mechanical noise in granular systems may modulate creep in an analogous manner to thermal fluctuations in glasses[34,35]. The precise role of mechanical noise remains somewhat ambiguous, and few experiments have been conducted to test whether disturbances are necessary for creep to occur—especially in the heap geometry[36,37].

In this study, we examine creep dynamics of an undisturbed subcritical sandpile, probing grain motions through time using an optical technique that allows us to observe exceedingly slow strain rates (Fig. 2). Experiments reveal that a creeping sandpile behaves like a relaxing glass. We also explore how disturbances can enhance or reverse aging. Results suggest that in the natural environment, hillslopes are made perpetually fragile by environmental perturbations. Comparisons of experimental creep profiles with data from natural hillslopes indicate that laboratory observations are generalizable.

## Results

**Undisturbed creep**. Our first objective is to demonstrate the existence of creep in a minimally disturbed model hillslope. Based on the previous work[23–27], we expect creep rates to be exceedingly slow ($\leq 10^{-6}$ m/s), which makes typical particle tracking methods impractical. Instead, we measure grain motions via spatially-resolved diffusing wave spectroscopy (DWS)[38], which determines strain associated with changes in the granular structure that occur on the order of the optical wavelength ($10^{-6}$ m; see "Methods" and Supplementary Note 3). Our experimental system consists of a granular heap initially prepared (time $t = 0$) just below the angle of repose, that is confined in an acrylic cell (Fig. 2) sitting on a vibration-isolating optical table (see "Methods" and Supplementary Fig. 2). Most experiments used glass beads with ideal optical properties; however, natural sand, kaolinite powder, and a mixture of the two were also tested (Supplementary Fig. 3).

The first result is that creep occurred for all experiments and granular materials, and it persisted over all observed timescales ($10^0 - 10^6$ s; Fig. 2, Supplementary Fig. 3, and Supplementary Movies 1–4). Initial creep velocities ($t = 0$), estimated from measured strain rates (see Supplementary Note 5), were on the order of nm/s (cm/year); i.e., comparable to measured rates of hillslope soil creep in the field (see below). All experiments exhibited glass-like "spatially heterogeneous dynamics"[32],

manifest as discrete, mesoscopic ($\gg d$) zones of strain that occurred throughout the system (Fig. 2b, c). At early times, these deformation zones were relatively larger and more concentrated near the sandpile surface. At later times, these zones became smaller and occurred less frequently, with lower spatial density. Cumulative strain $\epsilon$ resulting from this deformation diminished with depth beneath the surface because of increasing confining pressure, which restricts dilation that is often associated with grain rearrangement[23,27] (Fig. 1c). We also observed sensitivity to the preparation protocol, a ubiquitous phenomenon in fragile solids[39]. For example, the region of intense and persistent deformation seen near the pile apex (Fig. 2 and Supplementary Movie 1) always occurred at the location where avalanches had formed when the sand was first poured; this may be a particular feature of the preparation protocol[40].

Information on the time-dependent, global dynamics of creeping motion in the pile is encoded in the spatially-averaged correlation function of the speckle patterns, $\langle G \rangle$ (see "Methods"). $\langle G \rangle$ is in essence an autocorrelation function which charts how a configuration of grains (speckles) changes in time ($\tau$), with respect to a configuration at start time $t$. In the experiments reported here, $\langle G \rangle$ decayed monotonically with lag time $\tau$; this decay was most rapid at early times $t$ indicating fast grain motions, and slowed through time (Fig. 3). Normalizing the lag time of each correlation by the $e$-folding time, we find that the curves $\langle G \rangle (\tau/\tau_e)$ collapse onto a single exponential master curve (Fig. 3), consistent with previous observations of granular creep[23] and molecular dynamics simulations of glass[41]. The growth of the relaxation timescale $\tau_e$ increased as a power-law function of time (Fig. 3). Such power-law "aging" is a hallmark of creeping glass and other amorphous solids[35]. Our interpretation is that the initially loose sandpile has many "soft spots"[30] associated with low packing density and/or frictional contacts, and that strain relaxes these soft spots, redistributing stress within the system, leading to an overall slowing down of creep with time[35].

**The role of mechanical disturbances**. In the above description, granular creep progressively slows down. In this picture of relaxation, creep rates should tend asymptotically toward zero with time. Not all of our experiments, however, exhibited this behavior. Humidity fluctuations occurred for some runs, producing a complex response in terms of creep dynamics—notably at late start times (Supplementary Fig. 8). Data indicate that some

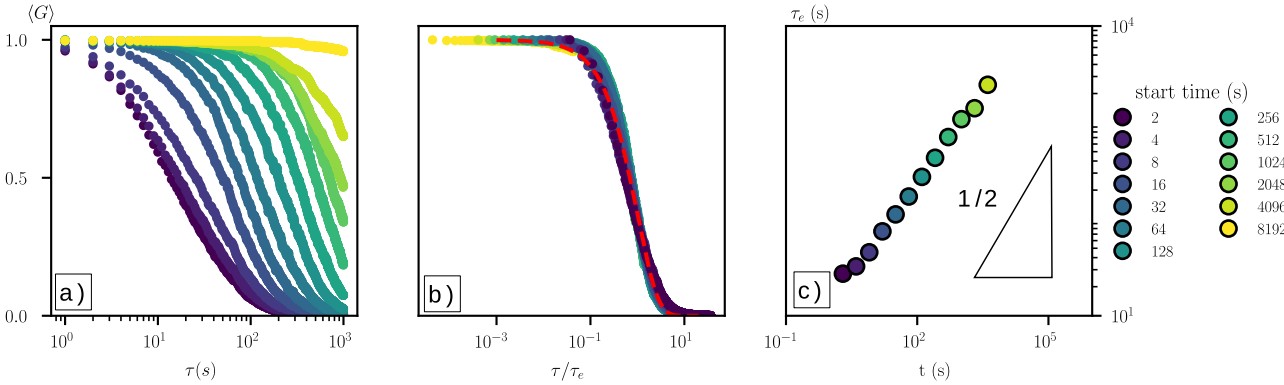

**Fig. 3 Glassy relaxation in an undisturbed granular heap. a** Spatially-averaged correlation function $\langle G \rangle$ for 13 start times. Start times mark the distance from the cessation of surface flow and the preparation of the pile. **b** Data are reasonably collapsed by $\tau_e$; red dotted line indicates exponential decay. **c** Growth of the relaxation timescale, with a slope of 1/2 shown for reference. Data may be fit with a least-squares regression of $\tau_e = \tau_{e_0} t^\theta$, where $\theta = 0.53$.

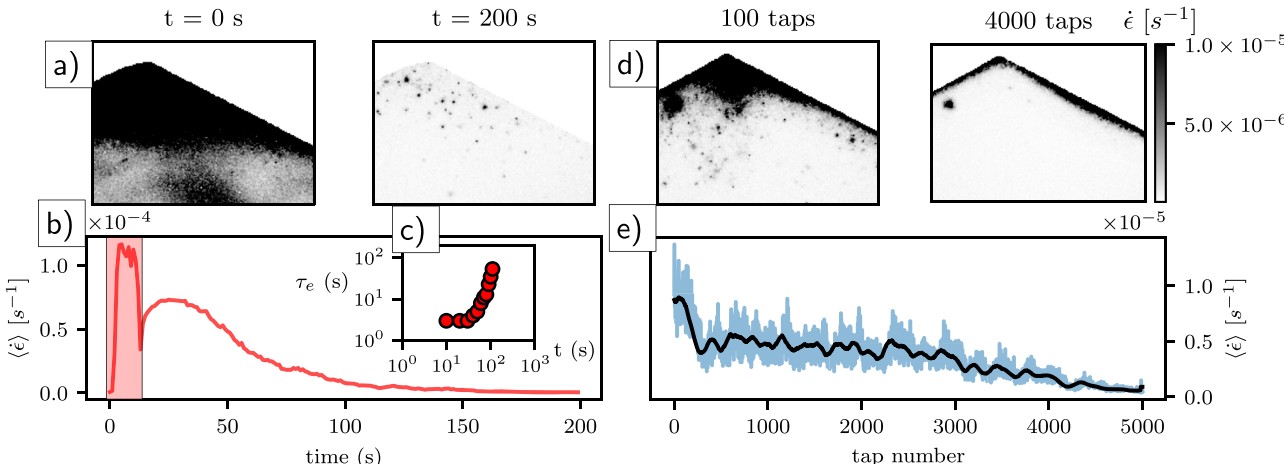

**Fig. 4 Mechanical perturbations drive rejuvenation or aging. a** Spatial maps of strain rate during and following heat pulse, with time indicated. **b** Spatially-averaged strain rate, including 10 s of heating applied (red rectangle), and relaxation after removal of the heat source. **c** Relaxation timescale $\tau_e$ during and following heat response (Supplementary Fig. 10). **d** Spatial maps of strain rate determined over one tap cycle (tap number indicated in figure). Note that after many taps, creep is mostly confined to a thin, localized layer at the surface. **e** Time series of spatially-averaged strain rate (blue line) during tapping (1 tap = 1 s). Black line indicates moving-window average (100 taps).

reversible (elastic) strain occurred in these runs—perhaps due to nanoscale capillary bridges or other tribological effects[42,43]. Similar behavior has been seen for weakly heated granular materials[38,44], suggesting that some kinds of disturbance may reverse relaxation and reactivate creep.

We posit that the same relaxation processes observed in our experiments also play out in natural soils, but that some environmental disturbances rejuvenate soil creep. Inspired by previous work[38,44,45], we examine heating as a method for creep rejuvenation in our experiments (see "Methods" and Supplementary Note 7). Thermal loading may be considered a proxy for shrink-swell daily temperature fluctuations that occur in natural soils[10–13]. The sandpile was first allowed to relax for $10^4$ s before applying disturbances. At the instant heat was turned on, an increase in strain rate $\dot{\epsilon}$ was observed as most of the pile began to creep faster (Fig. 4). This was likely due to heterogeneous thermomechanical stresses created by volumetric expansion of the grains[44], though expansion of the apparatus walls may have also played a role. Interestingly, the spatially-averaged strain rate $\langle \dot{\epsilon} \rangle$ (see "Methods") increased by more than ten times, reaching the same value observed at $t = 0$; i.e., just after preparation of the sandpile. Correlation functions also appeared similar to those observed at $t = 0$ (Supplementary Fig. 10). This demonstrates that a few seconds of heating was able to reverse $10^4$ s of aging. Once

heat was switched off, $\langle \dot{\epsilon} \rangle$ dropped immediately, then slowly decayed toward the preheating value (Fig. 4b). Repeated cycles of heating and cooling produced concurrent cycles of rejuvenation and relaxation, respectively; the overall effect was to sustain an approximately constant average creep rate, that did not decay with time (Supplementary Fig. 11).

Tapping of grains may induce surface flows on heaps, but also leads to compaction of the bulk[46,47]. Tapping may mimic some effects of seismic shaking of hillslopes[9]. We allowed an initial pile to relax for $10^4$ s, then tapped the pile with a metronome at 1 Hz (see "Methods"). Taps initially excited grains throughout the pile. As time progressed, however, a thin and fast-moving surface layer developed a sharp boundary at its base, below which the bulk grain motions slowed dramatically and became very intermittent (Fig. 4). The development of these two regimes is similar to the creep-flow transition observed in experiments[23] and simulations[24] of heap flows above the angle of repose (Supplementary Fig. 12). There was an overall trend of decreasing $\langle \dot{\epsilon} \rangle$ with increasing number of taps (Fig. 4). We conclude that vibrations fluidized surface grains but drove compaction in the bulk[47], leading to more rapid relaxation (compared to the undisturbed case) as the pile evolved toward a denser, lower-energy state (Supplementary Fig. 13). We interpret the boundary between fast and slow regions as a yield surface[27].

**Comparison with field observations.** Here, we examine previously published field data of horizontal ($x$) velocity profiles ($u$ ($z$)) determined from "Young pits" at four sites around the world, where creep was reportedly driven by different forcings[10–13] (Fig. 1). All velocity profiles are reasonably well described by an exponential function $u/u_0 = e^{-z/\lambda}$, where $z$ is depth below the surface, $u_0$ is the surface velocity, and $\lambda$ is a decay length determined from data fitting (Fig. 4c and Supplementary Fig. 6). The latter two parameters must be related to site-specific soil characteristics and environmental disturbance regimes, but exploring this is beyond the scope of this paper. Nonetheless, surface velocities for these hillslopes are of order $u_0 \sim 10^{-9}$ m/s (Fig. 4c and Supplementary Fig. 6), comparable to our estimated experimental creep rates for the initially loose and heated grains.

We compare our undisturbed creep experiments to field data, by first generating depth profiles of downslope ($x$)-averaged cumulative strain through time from the surface to 1-cm below (Fig. 1c). Our experiments measure strain rate rather than velocity (the latter may be crudely estimated, see Supplementary Note 5). The normalized strain rate profile $\dot{\epsilon}/\dot{\epsilon}_0 = e^{-z/\lambda}$, however, is essentially equivalent to a normalized velocity profile. We see that our experimental data fall on top of the field profiles (see Supplementary Fig. 3 for experiments with other materials). It is important to note, however, that while exponential profiles have been reported for granular creep in many experiments[23,25,26], an exponential profile is not diagnostic of creep. Also, creep in highly heterogeneous soils, or soils with macro-scale disturbances, such as tree throw[48], can exhibit erratic velocity profiles that are not well fit by an exponential. In such circumstances, macro-disturbances may locally and intermittently dominate downslope transport. Nonetheless, we posit that their primary role remains as a rejuvenating agent of perpetual, chronic creep; thus maintaining the fragile state of the hillslope.

## Discussion and outlook

By probing a seemingly static sandpile with speckle imaging, our experiments have revealed a seething and ceaseless creeping motion—even in the near absence of mechanical disturbances. These motions are strikingly similar to recent observations of creep in a heap of Brownian (micron-scale) particles[49], even though our sand grains are non-Brownian. Further, we have shown how granular creep rates can be tuned by imposing external disturbances that are geophysically relevant. Our experiments reveal deep similarities in how grains and glasses creep—both in the relaxation dynamics, and in the spatial correlations of strain (Supplementary Note 8)—which provide experimental evidence that mechanical disturbances in granular systems play a role akin to thermal fluctuations in glasses[31,34,35].

Intriguingly, even though the mechanics of grain motion are fundamentally different from Culling's soil creep model, our final result provides a kind of confirmation of his physical intuition[15]. In particular, heterogeneity in granular structure leads to seemingly random grain motions that decrease with depth; and mechanical disturbances can introduce new stresses and/or porosity that facilitate motion. Creep motions are consistent with granular self diffusion[50]; however, this does not imply there is any Culling-like diffusion relation between flux and slope. Moreover, Culling and subsequent hillslope researchers did not anticipate persistent creep even in the (near) absence of disturbance. Our system is intentionally prepared close to the critical state. Certainly, this means that creep rates are nearly as fast as they can be, and we expect them to slow exponentially with decreasing slope[24]. Examining behavior at lower slopes is beyond the scope of this paper, where our intent is to explore how and why creep is

happening at all—and how it is driven or suppressed by disturbance.

How do we understand the similarity in creep rates and profiles between our undisturbed and initially loose sandpile, and natural (disturbed) hillslope soils? Our new view separates creep into a generic relaxation process whose rate depends on granular friction/cohesion and structure, and diverse rejuvenation processes associated with environmental disturbances. We speculate that the primary role of (bio)physical disturbance in natural hillslopes is to maintain soil in a loose and fragile state, where relaxation rates are high. This is supported by our experiments, where temperature and humidity fluctuations reversed aging, and sustained high strain rates. Externally imposed shaking, however, can have the opposite effect. While vibrations in our experiments excited surficial flow, the underlying bulk became more rigid and less susceptible to future fluidization. This finding may have relevance for landslide development from earthquakes, and should be explored further.

Although soil is sensitive to disturbances, geologic history, and boundary effects[24,31], qualitative creep dynamics are robust across materials and environments. Future granular simulations could be used to reveal how disturbances influence the contact forces and/or structure that ultimately drive creep. Experiments could examine the consequences of cohesion/adhesion, surface charge, moisture, bioturbation, and other effects on creep dynamics. Resolving these factors will allow derivation of a coarse-grained creep rheology model, whose kinematics and scales are determined by physically meaningful parameters. Our results indicate that elastoplastic models developed to describe the rheology of amorphous solids[35]—that can explicitly incorporate mesoscopic scales of grain rearrangements, and rejuvenation by mechanical noise—may be good candidates. An initial investigation suggests that the granular system examined here has the essential elements of the elastoplastic worldview, in the form of a quadrupolar spatial correlation in the strain field (see Supplementary Note 8). An improved model of soil creep is not only useful for predicting hillslope sediment transport; it will also help us to better understand how creeping soil accelerates to the yield point, which leads to catastrophic landslides[24].

## Methods

**Measuring grain motion.** The principle of DWS is that highly coherent light illuminates our granular heap, where photons scatter and interfere, which produces a random "speckle pattern" that is collected with a CCD camera (Fig. 1 and Supplementary Note 3). As grains slowly creep past one another, they change the photon trajectories and render new speckle patterns. We achieve spatially-resolved measurements by partitioning images into a grid with cells (metapixels) of size $l^*$, the mean free path of photons within the material. This quantity is around $l^* \approx 3d$ for the granular materials used (see Supplementary Note 3). Fluctuations in the speckle pattern between a start time $t$ and a lag time $\tau$ within each metapixel are quantified via the normalized correlation function, $G(t, \tau)$ (Fig. 2)[38]. Global dynamics across the whole sandpile are measured by averaging $G$ for each metapixel, signified as $\langle G \rangle$. This allows determination of the first important quantity for assessing glassy dynamics: the relaxation time $\tau_e$, determined as the time at which $\langle G \rangle = 1/e$ (Fig. 2). For most experiments, we used monodisperse glass beads (Cerroglass), of diameter $d_s = 100$ μm and density $\rho = 2.6$ g/cc, to build the sandpile. This material was chosen because its scattering properties are well understood and it is standard in DWS experiments. From the correlation function $G$, we can apply optical theory[38] to determine the second important quantity for examining glassy dynamics: $\epsilon$, the strain that occurs within a volume set by $l^*$ (see Supplementary Note 3). Whereas DWS can still be used to examine relative grain motions for the sand and clay mixtures we used, use of more complex materials precludes us from calculating $l^*$, and hence from determining absolute strain $\epsilon$ (Supplementary Note 5).

**Experimental procedures.** Our experimental system is not meant to be a scaled model of a hillslope, either in a geometric or dynamic sense. Rather, it is designed to optimize the direct observation of grain motions, in order to understand the granular physics of creep that are relevant for soil motion at the pedon scale in nature. Reported experiments were conducted in relatively constant ambient temperature (21 °C ± 0.2) and relative humidity (23.8% + 0.3) conditions

(Supplementary Fig. 7). The heap was prepared by allowing a fixed volume/flow rate of granular material (well within the continuous-flow regime[51]) to flow out of a funnel, at a fixed height 8 cm above the center of the cell bottom. Results are reported for glass beads, unless otherwise stated. Our "undisturbed" experiments consisted of allowing the initial pile to relax under gravity, with no imposed external disturbances. We note, however, that small-scale ambient fluctuations in temperature and relative humidity did occur (Supplementary Fig. 7). We conducted a "short" duration experiment at a frame rate of $f = 1$ Hz for $10^4$ s (2.8 h) immediately following preparation, and a "long" duration experiment with $f = 0.2$ Hz for $10^6$ s (11 days). Image collection began at the start of emptying the funnel, while analysis of creep dynamics reported here started as the last grain entered the system and avalanching ceased ($t = 0$)—making the initial condition a sandpile prepared just below the angle of repose (Fig. 1). We computed both the instantaneous strain rate determined from successive image pairs through time, $\dot{\epsilon}(\tau = 1 \text{ s}) = \epsilon(t)f$ (e.g., Fig. 1), and the temporal evolution of the relaxation timescale $\tau_e$ sampled from different start times $t$, for each metapixel in an image. From these, we generated ensemble-average values for each image, $\langle \dot{\epsilon} \rangle$ and $\langle \tau_e \rangle$, that characterized the spatially-averaged dynamics of the sandpile through time (Figs. 1–3). From instantaneous strain values, we also computed surface-normal ($z$) profiles of downslope ($x$)-averaged strain (Supplementary Figs. 4 and 5); this allowed us to generate depth profiles of cumulative strain through time, for comparison to field data (Fig. 4).

**Disturbance protocols**. For experiments with disturbance, a pile prepared following the protocol above was allowed to relax for $10^4$ s before disturbances began. Heating of glass beads produces a small, but measurable volume expansion[44] (coefficient of thermal expansion $\sim 10^{-6}$ K$^{-1}$) that is reversed as grains cool. At $t = 0$, heat was applied to the side of the cell for 10 s by a heat gun, producing a measured sidewall temperature of 50 °C (Supplementary Fig. 9). After 10 s, the heating element was removed, while the creep response was documented for another 200 s (Fig. 3). For tapping experiments, discrete taps were delivered to the pile using a metronome (double pendulum) that rests on a platform attached to the cell (Supplementary Fig. 9). At $t = 0$, we initiated a series of 5000 taps delivered at a rate of 1 Hz, and recorded images at the same rate—but phase-lagged from the taps —for the 5000-s duration (Fig. 3 and Supplementary Fig. 9).

## Data availability
Codes to reproduce the results presented in this paper are hosted on this site: https://github.com/nakul-s-deshpande/fragile-hillslopes, where a link to the data may be found.

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

## Acknowledgements

We thank L. Galloway, A. Gunn, B. Ferdowsi, and D.J. Durian for helpful discussions. Research was supported by the Army Research Office (ARO W911NF-20-1-0113) and the National Science Foundation (UPenn MRSEC, DMR-1720530).

## Author contributions

N.S.D. performed the experiments and analysis; D.J.J. supervised the research. N.S.D., D.J.F., P.E.A., adn D.J.J. all contributed to interpretation and writing.

## Competing interests

The authors declare no competing interests.
