## [Peer Review File · Nature Communications]

Reviewers' comments:

Reviewer #1 (Remarks to the Author):

Summary

For the last century, hillslope geomorphologists have operated under the assumption that disturbances both biological (gopher digging, tree throw) and physical (freeze thaw, wetting drying) are necessary to drive hillslope soil transport. These "random" processes are thought to result in directed random motions of soil downhill, such that soil moves faster on steeper slopes according to a diffusion-like equation. However, due to the extremely slow nature of soil motions, it is difficult to observe hillslope sediment transport in real time, and even more difficult to recreate it meaningfully in a laboratory setting.

Deshpande et al. use cutting edge experimental technology to demonstrate, for the first time, that a granular pile can creep at rates comparable to observed soil velocities without any disturbance. Drawing from modern granular and soft matter physics, they convincingly show that soil likely behaves as a glass-like amorphous solid that experiences plastic deformation known as creep—importantly—just under the influence of gravity. This is a remarkable finding that will fundamentally change the way we think about hillslopes. Further, rather than suggesting that disturbances don't matter, they go on to show how various forms of disturbance change creep styles and rates. Heating can rejuvenate creep processes and allow them to continue indefinitely; tapping, on the other hand, results in a fast-flowing surface layer with slower creep rates at depth, where grains become compacted. This lays the foundation for modern hillslope geomorphology that draws connections between natural processes and the latest advances in materials science.

The importance and technical precision of this work are clear, and the supplementary material is transparent and sufficient for reproducibility. That said, I have some comments and questions, outlined below. Minor note: the main text frequently refers to the wrong figure, so the authors should make sure to double check all figure references.

Main comments/questions

Physical mechanism for creep:

Given the novelty of this work, I think the authors would be remiss to not include some sort of physical explanation for creep. I understand that this either 1) exists already in the granular creep literature 2) is not easy to explain 3) is not completely understood. However, the major impact of this paper—showing that soil can move essentially "by itself"—might be lost if readers finish reading without having some kind of understanding of the mechanism for creep. If I understand correctly, the qualitative explanation would be that grains are disordered, and therefore never perfectly organized or in a "happy" state. Therefore, gravity and frictional effects lead to creeping motions as the net force on each grain is not zero. Is there any kind of intuitive explanation you can give to help the reader along? I think it's essential to include something about this in the main text, but perhaps also in the supplement in the first section, after explaining why Culling's formulation is incorrect.

Surface slope of the granular heap:

A major point of this paper is that the classic hillslope diffusion "law," which related soil flux to topographic slope, is unfounded. However, there is little discussion about how surface slope might affect creep rates in granular materials. The authors note that experiments are prepared "just below the angle of repose." How might this change the strain rates they observe? I understand that particle motions at exceedingly slow speeds are difficult to measure (one of the most impressive aspects of this study), so perhaps it is too difficult to capture motions well below the angle of repose. A recent study (Ferdowsi et al., 2018) conducts simulations of granular creep and proposes that an increase in topographic slope may essentially increase local friction coefficients, thus leading to faster creep rates. Could this effect ever be observed in experiments, and how might it relate to predictions of soil transport in the field? To what extent might a lower slope slow creep rates- would it have a minimal effect, or potentially lower rates by orders of

magnitude? Given the short length of the paper, a sentence or two in the main text would be sufficient to address this point, along with some discussion in the supplement if the authors find it worthwhile.

Comparison with field velocities:

It's quite intriguing that the average velocities measured in these experiments are of the same order of magnitude as those measured on hillslopes. However, I also find it surprising, given the simplicity of the experiment (spherical glass beads, no cohesion) and the seemingly important role of disturbances in the field (frost heave, for example, can loft soil upward substantially) that must be present in the field velocity profiles shown in Figure 1. Assuming that cohesion and grain angularity would slow down creep rates—though I don't know that for a fact—could these effects, in conjunction with efficient disturbance in natural environments, lead to the similar observed rates? This also leads to another question that, while probably outside the scope of the paper, is relevant for its applicability to natural hillslopes: does the thickness of the granular pile make any difference in average creep rates? Observations of creeping soil displacement on natural hillslopes show motion (at least over human timescales) up to ~0.5 meter deep. Should we expect this increased depth compared to the experiment to result in faster rates?

I might also argue that the field profiles also don't measure true velocities, but accumulated displacement. For processes like freeze thaw, which often propagate from the surface downward into frozen (assumed to be immobile) soil, displacement changes with depth really represent time-integrated velocities and may be complicated by the fact that the surface has been "mobile" for a longer period of time. Do you think this effect may be happening in your experiment? In the supplement you mention that creep propagates downward from the surface through time. Does this have any bearing on the question of a possible depth-dependence of transport rates? It's probably not necessary to address these questions in the main text, but perhaps in the supplement, if the authors find they have something interesting to say about them.

Main text line-by-line comments:

Line 12: "visualize." maybe "examine" or "document" instead? This doesn't immediately sound like an experiment.

Line 16: "initially-fragile." No hyphen needed.

Line 41: change to "rather than sub-critically creeping"

Line 44: add "A flowing layer at the surface"

Line 50: "any particle configuration is metastable, and very small perturbations can lead to structural rearrangements." I think this is where you could describe the mechanism for creep more intuitively (see main comments above).

Figure 1 caption:

What type of environment is the photo from b found in?

What do the two different "soil creep" symbols correspond to?

Line 93: "relaxes" I understand, but "relaxes" may be confusing. maybe "hardens"?

Line 145: "seething and ceaseless." Very poetic.

Line 147: "even though our sand grains are non-Brownian." What is the significance of this?

Line 156: "this does not imply..." this is where you could potentially reference the Ferdowsi paper and discuss the possibility of a slope control on flux.

Line 171: "coarse-grained." I like the pun, but it might be confusing to the reader. Find another term that

doesn't include "grain"

Line 184: "see methods" this is the methods... maybe "see supplementary" instead?

Supplement Line-by-line comments

S1: I really love this section. I wonder if it might be more useful to the reader to add a paragraph explaining- not in detail, but in summary- the relevant physical interpretation for the creep experiments presented here. I understand this may not be completely understood, but is important to highlight the differences between this study and the Culling formulation.

Figure S1 caption: "Data indicate that all observed grains were in the dense granular flow, rather than creep, regime." even at depth? it's just hard to tell the order of magnitude for the points close to $I=0$.

S3 equation: define the "I" parameters.

Figure S4 caption: Would using vertical depth change the observed strain rate profiles from matching the field profiles?

Figure S6: What are the red lines?

Table 1: I believe the DWS profile value should be $\sim 10^{-9}$

-Rachel Glade

Reviewer #2 (Remarks to the Author):

The article focuses on creep in granular heaps by means of an experimental study with Diffusive Wave Spectroscopy (DWS) and proposes a direct analogy with natural hill slopes on the basis of the observation of (i) comparable strain rates and (ii) exponential creep profiles. More precisely, the authors were able to measure (slow to extremely slow) deformations in their model heaps over tens of hours, with an exponential decay in time, although their particles are athermal. These deformations mostly affect the surface of the heap and exhibit an exponential decay with depth. Furthermore, weak disturbances (through heating or tapping) are shown to rejuvenate the system and restore the initial regime of "faster" creep, thereby reversing the decay of creep rates (i.e., "aging"). Parallels are drawn with natural disturbances affecting soil.

There is no doubt that the search for well controlled model systems that mimic the natural dynamics of soil and for quantitative analogies between the latter and their model counterparts is worthy of interest. Furthermore, the manuscript is enjoyable to read on the whole and I think that the broad conclusions of the paper about the existence of (qualitative) similarities between granular creep, creep in thermal materials, and soil dynamics are trustworthy.

On the other hand, the central claim of the paper is not unprecedented (the aforementioned analogies can already be found in the literature, see e.g. Refs 29 and 31 of the manuscript) and therefore not entirely novel. Nor is the idea that perturbing an amorphous solid in various ways can rejuvenate it. Besides, the

arguments exposed in the manuscript to give a more quantitative footing to the analogies are relatively weak: They rest on the similarity of rather roughly evaluated strain rates and the observation of strain rates that decay exponentially with time and depth. Marked discrepancies are overlooked on the way, such as:

- the widely different decay lengths in granular heaps and natural soils (1mm=10 bead diameters vs. 10 cm in soils-- but what is the elementary length scale for soils?),
- the (acknowledged) fact that the exponential decay is not ubiquitous in nature,
- the fact that, under constant applied stress, creep in amorphous solids actually decays as a power law of time in most cases (power-law, or Andrade, creep) rather than exponentially, which may point to an effect of the heap geometry more than a material effect.

(Less importantly, aging in amorphous solids does not always affect the material properties algebraically; linear and logarithmic aging are also reported in some cases).

Therefore, some of the strong assertions exposed in the conclusion are not sufficiently bolstered by the data, e.g., "strikingly similar to recent observations of creep", "compelling evidence that mechanical disturbances play a role akin to thermal fluctuations".

On a more technical note, I also have some concerns about the evaluation of the strain rate from the DWS measurements, in particular about how the effect of the rattling motion of grains in the less compact superficial layers and of rigid translations (see technical details below).

All in all, while I am convinced that further experiments in this vein and more thorough analysis will lead to very interesting results, I do not believe that the results presented in this manuscript warrant publication in Nature Communications.

DETAILED COMMENTS

~~~~~

\* There is a problem with the figure numbers, which are shifted by one ("Fig. 1" should in fact be Fig. 2, I think)

#### INTRODUCTION

\* Some terms, which have a specific meaning for glasses and amorphous solids, are used loosely in the introduction, e.g., fragile (which has a very specific meaning for glasses) to allude to the marginal stability of the systems, or rigidity. I understand the need to use simple terms in the introduction, but would at least suggest to put these terms between quotation marks: fragile -> "fragile".

\* "in an analogous manner to thermal fluctuations in glasses. No experiments, however, have been conducted to test these ideas."

 I think that this assertion is not quite accurate. See, e.g.,

Pons, A., Amon, A., Darnige, T., Crassous, J., & Clément, E. (2015). Mechanical fluctuations suppress the threshold of soft-glassy solids: the secular drift scenario. *Physical Review E*, 92(2), 020201.

Reddy, K. A., Forterre, Y., & Pouliquen, O. (2011). Evidence of mechanically activated processes in slow granular flows. *Physical Review Letters*, 106(10), 108301.

#### UNDISTURBED CREEP

\* As indicated above, power-law creep has often been reported for amorphous solids subjected to uniform shear. This means that the effect of the heap geometry, which involves heterogeneous stresses, may have been understated in the manuscript.

#### MECHANICAL DISTURBANCES

\* Please note that the effect of thermal cycles on a granular pile has previously been studied by Divoux, T., Gayvallet, H., & Géminard, J. C. (2008). Creep motion of a granular pile induced by thermal cycling. *Physical review letters*, 101(14), 148303.

\* The analogy between the heating conducted in the study and freeze-thaw cycles in natural soils is dubious, as freezing also involves (potentially very strong) crystallisation forces that may break rocks.

#### COMPARISON WITH FIELD

\* The inference of velocities from scalar strain rates is not rigorous -- although it is most probably true with an exponentially decaying strain rate.

#### DISCUSSION

\* "Compelling evidence that mechanical disturbances play a role akin to thermal fluctuations in glasses"  While I agree qualitatively, I question the quantitative validity of this assertion (which is implied by "compelling evidence"): As far as I understand, the experiments do not provide quantitative insight into this issue, which is debated theoretically. See

Nicolas, A., Martens, K., & Barrat, J. L. (2014). Rheology of athermal amorphous solids: Revisiting simplified scenarios and the concept of mechanical noise temperature. *EPL (Europhysics Letters)*, 107(4), 44003.

Agoritsas, E., Bertin, E., Martens, K., & Barrat, J. L. (2015). On the relevance of disorder in athermal amorphous materials under shear. *The European Physical Journal E*, 38(7), 71.

#### DISTURBANCE PROTOCOLS

\* It might be worth stating that the heating protocol most probably induces a heterogeneous temperature field in the heap.

#### SUPPLEMENTARY

##### SECTION 1

\* I must admit that I cannot assess whether the authors' self-confessed partly 'Harry-Potterish' interpretation of Culling's theory is correct or not, but manifestly this sketchy description is rather far from the actual situation in a granular heap or in a soil, even schematically. In particular, the Brazilian nut effect and its kin should warn against the strict application of equilibrium concepts from hydrodynamics to sandpiles.

##### SECTION 2

\* The proposed hydrodynamic expression for the variations of pressure with depth is *\*not\** valid in a granular heap (see Janssen effect, force chains, etc.). Pressure under the apex may even be minimal.

##### SECTION 3

\* What is "tau"? Is it an absolute time or a lag time? In the latter case, shouldn't it be  $I(t+\tau)$  rather than  $I(\tau)$  in the equation?

\* **IMPORTANT TECHNICAL CONCERN:** I am concerned about the deduction of a strain value from the decay of  $G$ :

- how robust is this inference to rigid-body translations? In other words, does a rigid translation of the system lead to a decay of  $G$ ? In that case, if there were (for the sake of the argument) of a shear-band at some depth underneath the surface, the plain translation of the upper layer would be mistaken for strain, in the authors' interpretation, wouldn't it?

- As noted by the author, the compactness of the heap varies with the depth. Therefore, beads in the superficial layers may be more prone to rattling than their deep counterparts, which reduces  $G$  even in the *\*absence\** of strain. Unless I am mistaken, this is overlooked in the present interpretation, in which the decay of  $G$  is only associated with strain.

\* Fig. S3: shouldn't  $G$  always start at unity, at zero lag time? (So all curves on the right panels go to unity in their hidden parts on the left, at vanishing tau).

##### SECTION 5

\*  $u = 3 \times 10^{-9}$

 what is "x"? (It seems to be merely a "times" symbol)

\* The conversion between strain and velocities " $u = \dot{\epsilon} l$ " has shaky foundations: if the strain is pure shear strain, then it accumulates, so that the superficial velocity integrates the velocities of the layers underneath.

\* Fig. S9: Please correct the labels on the vertical axes.

Alexandre NICOLAS

Reviewer #3 (Remarks to the Author):

Deshpande and colleagues' study presents experimental evidence for sub-yield creep in an apparently static heap of grains with no external disturbances and below the angle of repose. The study's motivation is to explore internal dynamics leading to observed motion of grains down hillslopes. Classically, hillslope creep is attributed to external forcing, supplying the grains with energy and allowing them to move downslope even though the layer is below the angle of repose. The current contribution proposes that granular creep needs no external disturbances, but it is an emergent behavior due to the fragility of the grain skeleton, similar, in many aspects, to glass dynamics.

The declared motivation behind the study is somewhat transformative. The reason is that the necessity for external disturbances in inducing sediment motion down hillslope is the leading paradigm in the hillslope evolution literature, but it seems that no experiments, so far, were designed falsify it. Experimentally questioning this leading paradigm is what the current manuscript strives to achieve.

The manuscript employs a unique experimental setup, smart analysis, and it presents intriguing and unexpected results with potentially important implications. The manuscript, therefore, presents a significant contribution and could become influential within the landscape evolution, sediment mobilization, and soft matter communities. It was also a real pleasure reading it.

Below I raise some potentially critical issues regarding the implications of the results, but I believe that these could be addressed as part of an extended discussion, and they do not significantly hamper the importance of the discoveries. One way to address them might be to define better the domains in which applications are relevant.

I am not knowledgeable enough to comment on the experimental setup and immediate analysis, so my review focuses on the interpretation and implications of the presented results.

First, whether the authors' insights present a significant shift (or a fundamental change in views) in the leading paradigm is a delicate (or fragile) matter. The demonstrated creep-without-disturbances rapidly decay exponentially, and disturbances are needed to bring the strain rate back to its initial values that are representative of naturally observed creep rates over hillslopes. This means that external disturbances do matter to allow persistent, realistic creep rates. The question, therefore, is if disturbances felt by natural hillslope material only increase soil fragility and allow it to creep below yield, like in the experiments presented here, or if disturbances push the state of stress/energy beyond the yield envelop. I believe that we do not know the answer to that, but if the second is true, then deriving implications from the experimental results to natural settings is far from straightforward.

Second, the experiments are performed below the angle of repose, but importantly, very close to it. I.e., grains are at  $\theta_{\text{stop}}$ . Could it be that proximity to yield drives the observed glassy dynamics? If this is the case, then maybe shallower slopes (which are most common in natural terrains) would not show the same tendency to creep, or maybe the creep would be so much slower that it would not be a dominant effect in nature.

Finally, the implications of extrapolating creep kinematics over long timescales possibly deserve further discussion. A rate of cm/yr in both the experiments and in the field represents a relatively rapid landscape

evolution. Such terrains could be more prone to landsliding. In this case, creep motion, even when integrated through time, might be overwhelmed by the effect of episodic landslides in terms of both landscape evolution and sediment flux. On the other side of the spectrum, there are hillslopes that evolve at a much slower rate (clearly, field measurements were done where the rates are at the high end of the range), possibly due to lower slope and/or cohesion, and it is not clear how the insights derived from the experiments relate to them.

Some minor issues:

Line 34 + Section S1. The text in section S1 is really nice, but I think I missed the point. In what sense Culling's theory is inconsistent with granular mechanics? Or to ask it in differently, why was there a need to move from grains to air molecules and back to grains to understand Culling?

Related to this issue: 'Engorgio' would probably be the spell to turn molecules to giant grains.

Wrong Fig references in lines 108, 114, 136 & 122.

'Start time' is not immediately clear. Consider adding an explanation.

Line 163. The amplitude and frequency of external shaking dictate whether shaking tightens the packing or increases porosity. (BenDror and Goren, 2018 and many others).

Line 217: Citation bug.

Section S2 is very interesting to read.

Section S3: It is not clear how zeta, the random motion, is quantified, and as a consequence, how the strain could be validly extracted from  $\langle G \rangle$ ?

Liran Goren

# Responses to reviewer comments to Fragile Hillslopes

Nakul S. Deshpande1, David J. Furbish2,3, Paulo E. Arratia4, and Douglas J. Jerolmack1,4

1*Department of Earth and Environmental Science, University of Pennsylvania, Philadelphia, Pennsylvania, USA*

2*Departments of Earth and Environmental Sciences Vanderbilt University, Nashville, Tennessee, USA*

3*Civil and Environmental Engineering, Vanderbilt University, Nashville, Tennessee, USA*

4*Department of Mechanical Engineering & Applied Mechanics, University of Pennsylvania, Philadelphia, Pennsylvania, USA*

December 23, 2020

## 1 A case for revision and re-submission

We thank the editor and three reviewers for their thoughtful, deep and thorough reviews that, we believe, have greatly improved the quality and readability of our manuscript. We note that the reviews, while critical, were overall quite positive about our reported work. Moreover, we have been able to directly address the most critical points raised by the reviewers; therefore, we don't believe that the initial critiques 'sink' the paper. We have done our best to address the broad comments regarding the implications and applicability of our work apropos hillslope and landscape evolution. Further, we have worked on providing the reader with an intuitive sense of why the creep phenomenology we describe happens at all. Perhaps most important, we have directly addressed and mitigated the technical concerns raised by Reviewer 2 about the experimental setup. We have performed new analyses that convincingly demonstrate the veracity of the DWS technique. This validates our mapping from the correlation function to strain. In doing so, we find strain patterns that are similar to recently proposed (elastoplastic) models — including those published by Reviewer 2 — which bolsters the broader physical relevance of our experimental system, and its extension to natural hillslopes. We have addressed individual reviewer comments (in bold text) as follows and have highlighted our changes to the text (in red) below.

## 2 Reviewer 1 (Rachel Glade)

### 2.A Physical mechanism for creep:

Thank you for encouraging us to outline a physical intuition for why creep happens in the absence of disturbances; we overlooked this in the manuscript in light of the more technical aspects of the work. As you noted, this is in part due to ongoing debates at the theoretical level of the role of temperature vs. mechanical noise [1, 2] – the outcomes of which dictate how we outline a physical intuition. However, we speculate based on our qualitative observations that there are three basic ingredients for creep to occur: 1) disorder in the granular packing 2) the presence of walls and 3) an applied force. In this view, the irreversible, plastic microstrains associated with creep generate a kind of *internal noise* that allows creep to continue indefinitely [3]. This interpretation is also consistent with speculations from Ferdowsi et al. (2018) based on simulations of granular creep. Here we go further by adopting the physical picture outlined by the ‘elastoplastic’ framework. Imagine the group of  $\sim 10$  grains or so contained within each metapixel grid, in the maps of strain that we present. Because there is disorder, there is a distribution of local groups of grains that are able to yield. As this happens, stress is redistributed amongst neighboring grains, which pushes them further towards their own local yield conditions.

### 2.B Surface slope of the granular heap

Thank you for pointing to the role of the surface slope in affecting the dynamics we observe. Our choice of surface slope is in part due to the preparation protocol and the system geometry. An alternative system that allows for control of the surface slope has been examined separately in a similar system [4]. Based on these experimental results and the theoretical expectation, we hypothesize that the creep rates would decrease exponentially with slope below the critical point (angle of repose) – this is, as you noted, equivalent to increasing the effective friction as suggested by simulations [5] and consistent with the picture given by mean-field depinning [6].

### 2.C Comparison with field velocities

Thank you for encouraging us to strengthen our comparison with field velocities. We suspect that combinations of the physical conditions you outline may somehow cancel out. For example, a highly cohesive soil which is disturbed by strong temperature fluctuations may be able to creep at the same rate as a non-cohesive soil without strong disturbances. That is; the internal dynamics can be overridden by the externally imposed disturbances. We see your point about field displacement profiles not being the same as velocity profiles, where competing disturbances may only be active over certain portions of the soil column. We are not experts in these periglacial processes; moreover, we have been discouraged by those who are more familiar with these processes from lumping those systems in with ‘regular’ hillslope soil creep. However, we stand by the simplified stance that the long-term velocity is simply the displacement divided by the time of observation. This should be sufficient to describe an *average* velocity that incorporates intermittencies due to particular disturbances.

### 2.D line-by-line comments

1. **Line 12:** “visualize.” maybe “examine” or “document” instead? This doesn’t immediately sound like an experiment.  
changed text to “render experimentally”
2. **Line 16:** “initially-fragile.” No hyphen needed.  
hyphen removed
3. **Line 41:** change to “rather than sub-critically creeping”  
changed
4. **Line 44:** add “A flowing layer at the surface”  
added

5. **Line 50: “any particle configuration is metastable, and very small perturbations can lead to structural rearrangements.” I think this is where you could describe the mechanism for creep more intuitively (see main comments above).**  
We have added a sentence to emphasize that creep is essentially local yielding: **The origin of this metastability are locally weak zones which yield in response to the globally-applied stress.**
6. **Figure 1 caption: What type of environment is the photo from b found in?**  
added **“Sudetes Mountains”**
7. **Figure 1 caption: What do the two different “soil creep” symbols correspond to?**  
**We have modified the figure to indicate the location and corresponding reference of each dataset.**
8. **Line 93: “relaxes” I understand, but “relaxes” may be confusing. maybe “hardens”?**  
We feel that using the term “relaxes” is essential; though “hardens” is also appropriate, relaxation and rejuvenation are key descriptors for the phenomenology of glasses.
9. **Line 145: “seething and ceaseless.” Very poetic.**  
Thank you!
10. **Line 147: “even though our sand grains are non-Brownian.” What is the significance of this?**  
When particles are small ( $< 1\mu\text{m}$ ) and suspended in a fluid (air or water), they are susceptible to collisions with molecules within the fluid and undergo random ‘kicks’ as a result. This is Brownian motion [7]. If our particles were small enough, it is conceivable that the source of driving of creep comes from these thermal kicks. Indeed, experiments of heaps where the constituent grains are Brownian show similar creep and relaxation dynamics [8]. So, it is significant that our grains are non-Brownian because the nature of the driving/noise/energy which drives creep is more ambiguous.
11. **Line 156: “this does not imply...” this is where you could potentially reference the Ferdowsi paper and discuss the possibility of a slope control on flux.**  
We have elaborated on this point to emphasize that our pile is intentionally prepared near the critical point, where we expect creep to be as fast as they can be, and have referenced the Ferdowsi et al. work. We have added the following text: **Our system is intentionally prepared close to the critical state. Certainly, this means that creep rates are nearly as fast as they can be, and we expect them to slow exponentially with decreasing slope [5]. Examining behavior at lower slopes is beyond the scope of this paper, where our intent is to explore how and why creep is happening at all - and how it is driven or suppressed by disturbance.**
12. **Line 171: “coarse-grained.” I like the pun, but it might be confusing to the reader. Find another term that doesn’t include “grain”**  
We don’t deploy the term “coarsed-grained” as a pun – it is technical way to describe a mesoscopic lengthscale, one that encompasses a few grains but is still smaller than a bulk, continuum, macroscopic scale.
13. **Line 184: “see methods” this is the methods... maybe “see supplementary” instead?**  
**fixed**

## 2.E Supplement Line-by-line comments

1. **S1: I really love this section. I wonder if it might be more useful to the reader to add a paragraph explaining- not in detail, but in summary- the relevant physical interpretation for the creep experiments presented here. I understand this may not be completely understood, but is important to highlight the differences between this study and the Culling formulation.**  
In our view, the objective of the Culling section is to state that he had the wrong physics, and that the details of the physical interpretation of this experiment are in the Supplemental and the Main text. We have tried our best to make this intuition more clear in the text by highlighting the concept of

local yield vs. global yield but feel that additions to the Culling section would be distracting and/or redundant.

2. **Figure S1 caption:** “Data indicate that all observed grains were in the dense granular flow, rather than creep, regime.” even at depth? it’s just hard to tell the order of magnitude for the points close to  $I=0$ .

Adjusted caption - we make it more clear that all *observed* grain velocities are in the dense flow regime. Below this layer, no measurements were taken (it is presumed that they are motionless in the original study).

3. **S3 equation: define the “I” parameters.**

I see – this isn’t explicitly defined.  $I$  is the intensity field within the metapixel, not the inertial number, or another variable with “parameters”. We have made this definition more clear in the Supplemental Materials by adding: **where  $I$  indicates the metapixel intensity**

4. **Figure S4 caption: Would using vertical depth change the observed strain rate profiles from matching the field profiles?** The profiles should all be measured surface-normal. If measured otherwise, I imagine the profile would be distorted and may not match the field profiles (which are measured surface-normal)

5. **Figure S6: What are the red lines?**

They are the exponential fits. This is mentioned in the figure caption.

6. **Table 1: I believe the DWS profile value should be  $10^{-9}$ .**

No – the  $10^{-5}$  is correct. This is the difference between the strain-rate and the velocity.

### 3 Reviewer 2 (Alexander Nicolas)

We kindly thank you for your comments and depth of feedback. Our reading of your review indicates that the principal concern was the mapping from the correlation function to a strain. We note that there is a standard procedure described in the literature [4, 9, 10] for doing so, but understand that the Reviewer may still have reservations. To increase confidence that this mapping is correct – and that grains are not randomly ‘rattling’ in cages, but are instead responding to the globally imposed shear stress, we have addressed your comments re: Andrade creep and have engaged with your previous work within the elastoplastic framework. In addition to this main concern, we have responded to your comments below and made adjustments to the main text.

#### 3.A Introduction

1. ... **the central claim of the paper is not unprecedented (the aforementioned analogies can already be found in the literature, see e.g. Refs 29 and 31 of the manuscript) and therefore not entirely novel. Nor is the idea that perturbing an amorphous solid in various ways can rejuvenate it. Besides, the arguments exposed in the manuscript to give a more quantitative footing to the analogies are relatively weak: They rest on the similarity of rather roughly evaluated strain rates and the observation of strain rates that decay exponentially with time and depth.**

We acknowledge that there is a rich tradition investigating aging and rejuvenation in amorphous materials and substantial work elucidating the analogies and distinctions between thermal and mechanical noise. We have made adjustments to the manuscript to note this more clearly. Our goal in this paper is to make a transfer of concept *from soft matter physics to geomorphology* and demonstrate the connections between amorphous solids and the natural environment. This in itself is a novel contribution, as the current state of understanding and practice in geomorphology is rooted in an outdated, poorly implemented and experimentally unverified statistical mechanical formulation [11].

We believe our experiments and results have genuine elements of novelty; granular creep in the absence of a flowing layer at a finite surface slope has not been observed in experimental heap flows, nor is it anticipated by the most popular and successful granular  $\mu(I)$  rheology [12]. Previous work of the dynamics of granular heaps [13, 14] do not make a thorough examination of this creeping zone. We have seen no previous studies that show aging of a gravity-driven granular pile. Moreover, we have also not seen experimental tuning of aging and rejuvenation in a (athermal) granular system. We are encouraged and motivated by your own development of the elastoplastic framework, which we seek to make contact with in future experiments which are outside the scope of this work.

**Marked discrepancies are overlooked on the way, such as:**

- (i) **the widely different decay lengths in granular heaps and natural soils (1mm=10 bead diameters vs. 10 cm in soils– but what is the elementary length scale for soils?)**

Thank you for raising this important question. According to our data compilation, an intrinsic length-scale for natural soils can be extracted from the fitted exponential velocity profile. We note that this lengthscale  $\lambda$  is on the order of 10’s of cm, but the number of long-term records in the literature are too few to more rigorously constrain this value. We speculate that in the field, cohesion between soil grains and the formation of aggregates in clay particles play a role in modulating this lengthscale. We cannot prove this as of yet, but we note two anecdotal pieces of support: (i) aggregates in cohesive soils are often many millimeters in diameter, or more than an order of magnitude larger than the largest constituent grains; and (ii) in our experimental pile, the lengthscale became larger when kaolinite was added to sand.

- (ii) **the (acknowledged) fact that the exponential decay is not ubiquitous in nature**

Although we sought to be as complete as possible in our data compilation, the literature is poor in number of Young Pits that measure soil deformation over decadal timescales. Note that we excluded data from other environments that could not be defined as ‘pure’ hillslope creep (permafrost [15] and glacial till [16]) for clarity. Indeed, many field measurements show that at some depth, there is a

shear band. Making connections between homogeneous creep and localized shear banding is outside the scope of this work.

- (iii) **the fact that, under constant applied stress, creep in amorphous solids actually decays as a power law of time in most cases (power-law, or Andrade, creep) rather than exponentially, which may point to an effect of the heap geometry more than a material effect.**

Thank you for raising Andrade creep as a typical creep behavior in materials. Note that our results are actually fully consistent with this view (see Figure 1 below). The Reviewer indicates that, in our experiments, creep decays exponentially. This is a misunderstanding; our correlation of image intensity decays exponentially (which is also generic behavior in amorphous solids), but this is only used to estimate a relaxation timescale. The power-law growth of this relaxation timescale with time in Figure 3 is actually showing the same thing as a power-law decrease of strain rate with time. If we map the value of  $1/e \langle G \rangle$  to  $\epsilon$ , we can generate a relation in which the strain-rate decays as a power-law of time — just as the Reviewer anticipated from Andrade creep. Again, while people have reported such creep in different amorphous solids, we have never seen this demonstrated for gravitational relaxation of a granular pile.

- (iv) **Less importantly, aging in amorphous solids does not always affect the material properties algebraically; linear and logarithmic aging are also reported in some cases.**

Thank you for noting that varying functional forms dictate the aging. Our results indicate power-law aging (see Figure 1 below).

**Figure 1: Decay of the creep rate.** We use the growth of the relaxation timescale (see Figure 3c of the main text) to compute a characteristic creep rate for each start time. This is done by simply mapping the value of  $1/e \langle G \rangle$  to a strain  $\epsilon$  and dividing the by timescale  $\tau_{efold}$ . The power of the fitted line is about 1/2; and consistent with power-law aging.

1. **There is a problem with the figure numbers, which are shifted by one ("Fig. 1" should in fact be Fig. 2, I think)**

Thank you for noting this error; we have fixed the figure references in the text so that they call out the correct figure numbers.

2. **Some terms, which have a specific meaning for glasses and amorphous solids, are used loosely in the introduction, e.g., fragile (which has a very specific meaning for glasses) to**

**allude to the marginal stability of the systems, or rigidity. I understand the need to use simple terms in the introduction, but would at least suggest to put these terms between quotation marks: fragile - "fragile".**

We are aware that the traditional meaning of ‘fragility’ in the context of supercooled liquids refers to a non-Arrhenius temperature dependence [17, 18]. However, there are other ways that ‘fragile’ is used in a soft matter context. For example, ‘fragile’ is also meant to describe the development of network fabric structure in shear-jammed granular materials [19–21]); we thus believe that ‘fragile’ is a term wholly appropriate in describing both phenomenologies. Here, we adopt a third definition, outlined in a recent review [22], embraces both the traditional terms, and opens the possibility for further contact between these definitions as defined in physics, and their applicability in geoscience settings.

3. **“in an analogous manner to thermal fluctuations in glasses. No experiments, however, have been conducted to test these ideas.” I think that this assertion is not quite accurate. See, e.g.,**

**Pons, A., Amon, A., Darnige, T., Crassous, J., Clément, E. (2015). Mechanical fluctuations suppress the threshold of soft-glassy solids: the secular drift scenario. *Physical Review E*, 92(2), 020201.**

**Reddy, K. A., Forterre, Y., Pouliquen, O. (2011). Evidence of mechanically activated processes in slow granular flows. *Physical Review Letters*, 106(10), 108301.**

Thank you for bringing these references to our attention; we are aware of these works and especially of Axelle Amon’s group work. We have made an adjustment to the text to state that **“few experiments in a heap geometry to test these ideas exist”**.

4. **As indicated above, power-law creep has often been reported for amorphous solids subjected to uniform shear. This means that the effect of the heap geometry, which involves heterogeneous stresses, may have been understated in the manuscript.**

Actually, our results are consistent with power-law creep. Our demonstration of the growth of the relaxation timescale is equivalent to the power-law decay of the creep rate. We note that this uniform shear is somewhat ambiguous; in the case of a flowing layer a global shear is imposed but here, only the shape of the pile imposes the shear, which isn’t a very satisfying control parameter because it doesn’t change.

### 3.A.1 Mechanical Disturbances

1. **Please note that the effect of thermal cycles on a granular pile has previously been studied by: Divoux, T., Gayvallet, H., Gémard, J. C. (2008). Creep motion of a granular pile induced by thermal cycling. *Physical review letters*, 101(14), 148303.**

Thank you for bringing this reference to our attention; we have included and discussed this in the main text. We note that this is a tube geometry and not a heap with a finite global shear stress. This work also interpreted their results as aging rather than rejuvenation – contrary to our results. We look forward to exploring temperature fluctuations in subsequent work.

2. **The analogy between the heating conducted in the study and freeze-thaw cycles in natural soils is dubious, as freezing also involves (potentially very strong) crystallisation forces that may break rocks.**

Thank you for this insight. We agree - the analogy is maybe too loose. Nonetheless, the conceptual approach of relaxation and rejuvenation via small temperature cycles is novel. We have changed the text to reflect this: **Thermal loading may be considered a proxy for shrink-swell daily temperature fluctuations that occur in natural soils.**

### 3.B Comparison with field

1. **The inference of velocities from scalar strain rates is not rigorous – although it is most probably true with an exponentially decaying strain rate.**

Indeed, we make this assumption because the strain rate profiles are exponential (we recognize that the integral of an exponential is itself an exponential).

### 3.C Conclusion

1. **”Compelling evidence that mechanical disturbances play a role akin to thermal fluctuations in glasses”** While I agree qualitatively, I question the quantitative validity of this assertion (which is implied by ”compelling evidence”): As far as I understand, the experiments do not provide quantitative insight into this issue, which is debated theoretically. See Nicolas, A., Martens, K., Barrat, J. L. (2014). Rheology of athermal amorphous solids: Revisiting simplified scenarios and the concept of mechanical noise temperature. *EPL (Europhysics Letters)*, 107(4), 44003. Agoritsas, E., Bertin, E., Martens, K., Barrat, J. L. (2015). On the relevance of disorder in athermal amorphous materials under shear. *The European Physical Journal E*, 38(7), 71.

Thank you for your comment. We have adjusted the claim and now state “experimental evidence” rather than “compelling evidence” and have made the point that our results are more qualitative than quantitative. We note that our identification of a quadrupolar shape to the spatial correlation of the strain field (see below) which we feel makes this connection more compelling.

### 3.D Disturbance protocols

1. **It might be worth stating that the heating protocol most probably induces a heterogeneous temperature field in the heap.**

Noted. Thank you for the suggestion. We have added **This was likely due to heterogeneous thermo-mechanical stresses created by volumetric expansion of the grains.**

### 3.E Supplementary

1. **Section 1. I must admit that I cannot assess whether the authors’ self-confessed partly ‘Harry-Potterish’ interpretation of Culling’s theory is correct or not, but manifestly this sketchy description is rather far from the actual situation in a granular heap or in a soil, even schematically. In particular, the Brazilian nut effect and its kin should warn against the strict application of equilibrium concepts from hydrodynamics to sandpiles.**

We agree with you wholeheartedly that Culling’s theory is sketchy. In conversations with colleagues in geomorphology and in previous reviews, we are often confronted with objections that center on the interpretation of Culling’s theory. The intent of this colloquial description is meant to outline a kind of physical intuition to demonstrate why it is wrong, and that our experiments should provide a phenomenological replacement for Culling. Indeed, even with this interpretation showing how unphysical Culling’s theory is, another Reviewer did not understand why the Culling description is not appropriate.

2. **Section 2. The proposed hydrodynamic expression for the variations of pressure with depth is *\*not\** valid in a granular heap (see Janssen effect, force chains, etc.). Pressure under the apex may even be minimal.**

Thank you for your comment about the application of our hydrodynamic expression. We are aware of the Janssen effect, force chains and the like – which are rarely taken into account in rheological descriptions (for example  $\mu(I)$ ). The hydrodynamic assumption is valid for a certain depth beneath the surface. We are also aware of the ‘dip under pile’ phenomenology and reference work examining this in the text — and note it in the persistent deformations near the apex, in the interpretation of the strain rate maps. Note that where we calculate the profiles (where we need a pressure term to calculate  $I$ ), is far from the apex of the pile. Also, this exercise is mostly meant to simply demonstrate that  $I$  is very small. Finally, in laboratory experiments with similar ratio of grain size/width [23], and in Discrete Element Method simulations of a granular heap [5], we were able to apply  $\mu(I)$  rheology. The Janssen effect should become relevant for grains deeper below the surface; we did not examine those grains. In a previous paper we provided a justification for neglecting the Janssen effect for a similar experimental geometry [24].

3. **Section 3. What is ”tau”? Is it an absolute time or a lag time? In the latter case,**

shouldn't it be  $I(t+\tau)$  rather than  $I(\tau)$  in the equation?

Yes, the equation should read with ' $t+\tau$ '. **fixed**.

4. **Fig. S3: shouldn't G always start at unity, at zero lag time? (So all curves on the right panels go to unity in their hidden parts on the left, at vanishing tau)**

Yes, each correlation function starts at unity, but because they are plotted by start time, dots for early start times are covered up.

5. **Section 5.  $u = 3 \times 10^{-9} \dot{\epsilon}$  what is "x"? (It seems to be merely a "times" symbol)**

Yes, this is a times symbol and **have un-italicized the 'x'**.

6. **The conversion between strain and velocities " $u = \dot{\epsilon} l$ " has shaky foundations: if the strain is pure shear strain, then it accumulates, so that the superficial velocity integrates the velocities of the layers underneath.**

This is only done for illustration purposes, to get a rough idea of velocity. Our 'instantaneous' strain maps are actually maps of the strain measured in each metapixel over a pair of images - that are one second apart. They are not cumulative strain maps (although those can, and have, been created as well).

7. **\* Fig. S9: Please correct the labels on the vertical axes.**

These are correct - degrees C and  $m/s^2$

### 3.F Important technical concern

**I am concerned about the deduction of a strain value from the decay of G: - how robust is this inference to rigid-body translations? In other words, does a rigid translation of the system lead to a decay of G? In that case, if there were (for the sake of the argument) of a shear-band at some depth underneath the surface, the plain translation of the upper layer would be mistaken for strain, in the authors' interpretation, wouldn't it? - As noted by the author, the compactness of the heap varies with the depth. Therefore, beads in the superficial layers may be more prone to rattling than their deep counterparts, which reduces G even in the \*absence\* of strain. Unless I am mistaken, this is overlooked in the present interpretation, in which the decay of G is only associated with strain.**

We have carried out a new analysis of the data, in response to your critique here. We have computed the 2D spatial correlation function, following procedures published in your work, and found it to be quadrupolar — consistent with the picture of Eshelby inclusions within an elastic medium — as found in simulations [25, 26]. Although tentative in terms of evidence, the quadrupolar pattern is strong across measured experiments. We have added a plot in the Supplementary Materials showing this, and plan to pursue this further in future work. We feel that this further justifies the physical mapping from G to strain. In the case of rattlers, we would not expect such structure to the spatial correlations.

If there were a shear band at the surface, there would be no strain rate profile within the bulk. Similar objections were raised in our previous works on creep, where reviewers suggested that some solid body slip of the bulk — rather than creep within the bulk — might be occurring. The fact is, that creep in our free-surface granular materials — from the fluid driven annular flume [24] to the simulated granular heap [5] to our experimental heap here — all show an exponentially decreasing speed with distance below the surface. The author is correct that pure translation would also cause decorrelation in our measurement. But, if particles were moving in solid-body sliding then they would all show the same apparent strain — not an exponential profile.

Note that in other granular DWS studies [27] the creation of shear bands has been analyzed; in other words, the technique doesn't have a problem measuring strain within the bulk and also identifying strain along the band itself. In previous studies of a free-surface granular pile, an apparent shear band emerged as the pile was tilted to the angle of repose — i.e., it heralded failure in the bulk [4]. Our experiment remains below the angle of repose so no shear band arises. Moreover, in our previous work in which grains were driven above critical, we observed a *fluidized* surface layer transitioning, at depth, to a creeping regime [5, 23]. We see no reason why this system would not do the same if driven above critical, something we intend to test in

the near future. Nevertheless, based on the spatial heterogeneity of localized strain, the exponential strain profile when averaged over all of this heterogeneity, and the quadrupolar nature of the strain, demonstrate that these displacements are not the result of a solid-body translation or a shear band.

**Figure 2: Quadrupoles at three times.** These are the 2d spatial autocorrelation functions for three different timesteps within our creep experiments.

### 3.G Note to editor

We conducted this new analysis in response to Alexander Nicolas (Reviewer 2). We have added this figure to the supplemental material; below is the full text that we include there. Further, we add reference to this additional supplemental material in the main text in the final paragraph: “Our experiments reveal deep similarities in how grains and glasses creep – both in the relaxation dynamics, and in the spatial correlations of strain (Supplemental Materials Section S8) – which provide experimental evidence that mechanical disturbances in granular systems play a role akin to thermal fluctuations in glasses” and “An initial investigation suggests that the granular system investigated here has the essential elements of the elastoplastic worldview in the form of a quadrupolar spatial correlation in the strain field (see Supplementary Materials Section S8).”.

#### Section S8: Spatial correlation of the strain field

In many athermal materials, there is a characteristic quadrupolar spatial signature of the dynamics that result plastic flow events are triggered and stress is locally redistributed [25, 28–31]. To demonstrate that the decorrelations in our system are due to strains and the connections between this strain and those other amorphous solids, we compute the spatial auto-correlation of the instantaneous strain rate fields at three times (Fig. S 14). Practically, we compute this by taking the 2D Fourier transform of a square region of interest of the strain-rate field, reverse the transform and divide by the variance to normalize; this is the quantity  $C$ . Though this result is preliminary and the subject of future works, we include it here to emphasize the veracity of our mapping from  $G$  to  $\epsilon$  and to point out further similarities between our system and the deformation and flow of other amorphous athermal materials. Exploring these similarities more rigorously is outside the scope of this work but we look forward to pursuing these questions in the future.

## 4 Reviewer 3 (Liran Goren)

### 4.A Three main points

1. **The question, therefore, is if disturbances felt by natural hillslope material only increase soil fragility and allow it to creep below yield, like in the experiments presented here, or if disturbances push the state of stress/energy beyond the yield envelope.** In our view, the role of disturbances can be twofold: to either move the hillslopes closer to (rejuvenate) or farther from (age) the critical point. If the nature and magnitude of the disturbances are ‘large enough’ - we expect them to push the system to failure. For example, if we chose a larger vibration amplitude, we would expect to fully fluidize the system. We are certain that this is what happened in Roering’s classic experiments of acoustically disturbed granular pile, as laid out in the Supplemental. And of course this happens with landslides. The important idea is that plastic strain during creep IS failure — but it is localized failure. One could extrapolate from this that, should disturbances cause plastic rearrangements to percolate across the sample, we would get failure in the bulk. Indeed, this is the plastic depinning model described in our previous work [5] examining the creep-to-flow transition. In this paper we do not examine the flow part.
2. **Second, the experiments are performed below the angle of repose, but importantly, very close to it. I.e., grains are at  $\theta_{stop}$ . Could it be that proximity to yield drives the observed glassy dynamics?**  
Yes, absolutely. As articulated in [5] the dynamics vary exponentially in the vicinity of the critical point.
3. **Finally, the implications of extrapolating creep kinematics over long timescales possibly deserve further discussion.** We see your point, but this problem is no different than say, extrapolating bed-load flux to long timescales – the premise of the exercise is fraught with uncertainties and the underlying stochastic nature of the dynamics. It does seem that if creep persists indefinitely on hillslopes, then it reflects some kind of steady state; otherwise it would grind to a halt or runaway in landslide failure. Our work can help to explain *how* creep might persist in nature (rather than just aging and slowing down); but we are at the very beginning of this new understanding of granular creep.

### 4.B some minor issues

1. **Line 34 + Section S1. The text in section S1 is really nice, but I think I missed the point. In what sense Culling’s theory is inconsistent with granular mechanics? Or to ask it in differently, why was there a need to move from grains to air molecules and back to grains to understand Culling?**  
The point is that Culling imported the statistical mechanics of the day to describe grains. Nowadays, this is the realm of granular kinetic theory and has moved much farther than Culling’s understanding. First thing: Culling’s description is closest to a granular gas — a dilute or rarefied transport where grains are strongly driven and inertial collisions are dominant. So it is important to note that creep is at the other end of the spectrum: it is a dense amorphous solid, with enduring frictional contacts and no inertia. Second, Culling’s description really describes a gas like air, with then some strange interactions added. We want to make the point that, in actuality, his physical description does not apply to any material. Since the physical picture is incorrect, any mathematical formulation predicated on that picture is inadequate. We think it’s important to show this; it is not meant to disrespect Culling who was surely a pioneer, but rather to examine the formulation in hindsight and show how it is inconsistent with modern understanding. We propose that the proper framework is glassiness, and understanding deformation in amorphous solids. Nonetheless, the antiquated Culling perspective underlies contemporary geomorphology’s treatment of the soil creep problem.
2. **Wrong Fig references in lines 108, 114, 136 122.**  
references fixed

3. **'Start time' is not immediately clear. Consider adding an explanation.**  
We have added the explanation “Start times mark the distance from the cessation of surface flow and the preparation of the pile.” in the caption to Figure 3.
4. **Line 163. The amplitude and frequency of external shaking dictate whether shaking tightens the packing or increases porosity. (BenDror and Goren, 2018 and many others)**  
Thank you for making us aware of this reference. Indeed, even a system tapped at the same amplitude and frequency can oscillate between two packing states [32]. The situation here is beyond the scope of this paper but in subsequent work we plan to more fully explore the effect of shaking amplitude and frequency.
5. **Line 217: Citation bug.**  
bug fixed
6. **Section S3: It is not clear how zeta, the random motion, is quantified, and as a consequence, how the strain could be validly extracted from  $G$  ?**  
We do not explicitly account for zeta. Our guess is that it is a leftover from the original formulation to account for non-strain aligned motions [9] but have not come across a method to deconvolve it. Nonetheless, we are confident that our mapping from  $G$  to  $\epsilon$  reflects a true strain - see above in the response to Reviewer 2 and in the new Supplementary Methods Section S8.

## References

1. Agoritsas, E., Bertin, E., Martens, K. & Barrat, J. L. On the Relevance of Disorder in Athermal Amorphous Materials under Shear. *European Physical Journal E* **38**, 1–22 (2015).
2. Nicolas, A., Martens, K. & Barrat, J. L. Rheology of Athermal Amorphous Solids: Revisiting Simplified Scenarios and the Concept of Mechanical Noise Temperature. *Epl* **107** (2014).
3. DeGiuli, E. & Wyart, M. Friction Law and Hysteresis in Granular Materials (2017).
4. Amon, A., Bertoni, R. & Crassous, J. Experimental Investigation of Plastic Deformations before a Granular Avalanche. *Physical Review E - Statistical, Nonlinear, and Soft Matter Physics* **87**, 1–12 (2013).
5. Ferdowsi, B., Ortiz, C. P. & Jerolmack, D. J. Glassy Dynamics of Landscape Evolution. *Proceedings of the National Academy of Sciences* **115**, 4827–4832 (2018).
6. Agoritsas, E., García-García, R., Lecomte, V., Truskinovsky, L. & Vandembroucq, D. *Driven Interfaces: From Flow to Creep Through Model Reduction* (Springer US, 2016).
7. Einstein, A. On the Motion of Small Particles Suspended in a Stationary Liquid, as Required by the Molecular Kinetic Theory of Heat. *Annalen der Physik* **322**, 549–560 (1905).
8. Bérut, A., Pouliquen, O. & Forterre, Y. Brownian Granular Flows Down Heaps. *Physical Review Letters* **123**, 1–5 (2019).
9. Amon, A., Mikhailovskaya, A. & Crassous, J. Spatially Resolved Measurements of Micro-Deformations in Granular Materials Using Diffusing Wave Spectroscopy. *Review of Scientific Instruments* **88** (2017).
10. Amon, A., Nguyen, V. B., Bruand, A., Crassous, J. & Clément, E. Hot Spots in an Athermal System. *Physical Review Letters* **108**, 1–5 (2012).
11. Culling, W. E. H. Soil Creep and the Development of Hillside Slopes. *The Journal of Geology* **71**, 127–161 (1963).
12. Jop, P., Forterre, Y. & Pouliquen, O. A Constitutive Law for Dense Granular Flows. *Nature* **441**, 727–730 (2006).
13. Komatsu, T. S., Inagaki, S., Nakagawa, N. & Nasuno, S. Creep Motion in a Granular Pile Exhibiting Steady Surface Flow. *Physical review letters* **86**, 1757 (2001).
14. Katsuragi, H., Abate, A. R. & Durian, D. J. Jamming and Growth of Dynamical Heterogeneities versus Depth for Granular Heap Flow. *Soft Matter* **6**, 3023–3029 (2010).

15. Harris, C., Davies, M. C. & Rea, B. R. Gelifluction: Viscous Flow or Plastic Creep? *Earth Surface Processes and Landforms* **28**, 1289–1301 (2003).
16. Boulton, G. S. & Dobbie, K. E. Slow Flow of Granular Aggregates: The Deformation of Sediments beneath Glaciers. *Philosophical Transactions of the Royal Society A: Mathematical, Physical and Engineering Sciences* **356**, 2713–2745 (1998).
17. Ediger, M. D. Spatially Heterogeneous Dynamics in Supercooled Liquids. *Annual Review of Physical Chemistry* **51**, 99–128 (2000).
18. *Nonexponential Relaxations in Strong and Fragile Glass Formers: The Journal of Chemical Physics: Vol 99, No 5* <https://aip.scitation.org/doi/abs/10.1063/1.466117>.
19. Cates, M. E., Wittmer, J. P., Bouchaud, J. P. & Claudin, P. Jamming, Force Chains, and Fragile Matter. *Physical Review Letters* **81**, 1841–1844 (1998).
20. Berzi, D., T. Jenkins, J. & Richard, P. Erodible, Granular Beds Are Fragile. en. *Soft Matter* **15**, 7173–7178 (2019).
21. Zhao, Y., Barés, J., Zheng, H., Socolar, J. E. S. & Behringer, R. P. Shear-Jammed, Fragile, and Steady States in Homogeneously Strained Granular Materials. en. *Physical Review Letters* **123**, 158001 (2019).
22. Jerolmack, D. J. & Daniels, K. E. Viewing Earth’s Surface as a Soft-Matter Landscape. *Nature Reviews Physics* **1**, 716–730 (2019).
23. Houssais, M., Ortiz, C. P., Durian, D. J. & Jerolmack, D. J. Rheology of Sediment Transported by a Laminar Flow. *Physical Review E* **94**, 062609 (2016).
24. Houssais, M., Ortiz, C. P., Durian, D. J. & Jerolmack, D. J. Onset of Sediment Transport Is a Continuous Transition Driven by Fluid Shear and Granular Creep. *Nature Communications* **6**, 1–8 (2015).
25. Nicolas, A., Rottler, J. & Barrat, J.-L. Spatiotemporal Correlations between Plastic Events in the Shear Flow of Athermal Amorphous Solids. en. *The European Physical Journal E* **37**, 50 (2014).
26. Puosi, F., Rottler, J. & Barrat, J. L. Plastic Response and Correlations in Athermally Sheared Amorphous Solids. *Physical Review E* **94**, 1–6 (2016).
27. Houdoux, D., Nguyen, T. B., Amon, A. & Crassous, J. Plastic Flow and Localization in an Amorphous Material: Experimental Interpretation of the Fluidity. *Physical Review E* **98**, 1–8 (2018).
28. Chattoraj, J. & Lemaitre, A. Elastic Signature of Flow Events in Supercooled Liquids Under Shear. *Physical Review Letters* **111**, 066001 (2013).
29. Chikkadi, V., Wegdam, G., Bonn, D., Nienhuis, B. & Schall, P. Long-Range Strain Correlations in Sheared Colloidal Glasses. *Physical Review Letters* **107**, 198303 (2011).
30. Jensen, K. E., Weitz, D. A. & Spaepen, F. Local Shear Transformations in Deformed and Quiescent Hard-Sphere Colloidal Glasses. en. *Physical Review E* **90**, 042305 (2014).
31. Desmond, K. W. & Weeks, E. R. Measurement of Stress Redistribution in Flowing Emulsions. *Physical Review Letters* **115**, 098302 (2015).
32. Dijkstra, J. A. & Van Hecke, M. The Role of Tap Duration for the Steady-State Density of Vibrated Granular Media. *Europhysics Letters* **88**, 1–6 (2009).

## REVIEWER COMMENTS

### Reviewer #1 (Remarks to the Author):

In my view, the authors have succeeded in addressing my initial relatively minor comments on this paper, and have put considerable work into addressing other reviewers' technical concerns. The paper is well written, represents a fundamental advance in both geomorphology and materials science, and will be of great interest to readers of Nature Communications. I have no revisions to suggest, and would be happy to see this published as is.

### Reviewer #2 (Remarks to the Author):

That the authors have taken great care to address the Reviewers' concerns in depth is greatly appreciated. (Besides, I am flattered that the authors mention that they have engaged with my previous work, but I hope that this sentiment does not interfere with my review.)

The (new) computation of the spatial correlations of their G function abates my major technical concern regarding the validity of their experimental setup. For sure, it does not fully annihilate this concern, because it is not entirely clear to me how accurate ("quantitative") the correspondence between G and the local strain is, but at least it successfully evinces a relation between the two variables.

Another issue that I had was in keeping with how compelling is the experimental demonstration of the 'mapping' between a soft matter problem (granular creep) and an ongoing geomorphology issue (soil creep). The previous version was very assertive in this regard. The more nuanced wording of the revised manuscript sounds much more acceptable to me and more reflective of the present findings. In addition, this toning down of the claims does not make the manuscript any less enjoyable to read.

Clearly, the manuscript brings interesting pieces of evidence in favour of the idea of such a 'mapping' between the two problems, but I side with Referee 3 in that it does not bring the debate to an end: Even if one accepts that external disturbances may rejuvenate the soil and thus maintain creep at non-vanishing level, one may wonder if the immediate effect of these external disturbances (thermal effects, freeze/thaw, tree throw, etc.) may not prevail over the subsequent, more persistent creep that they induce, as far as displacements are concerned.

Still, I agree that this vast debate can likely not be fully solved in a single paper and, therefore, I see no more objections to the publication of the manuscript in Nature Communications, as it is now (except for the minor revisions advocated below). My opinion is that researchers from the soft matter community might be more impressed by the sophisticated technical setup than surprised at the reported findings. However, the other reviewers have suggested that the echo may be much stronger (in terms of novelty) in the community investigating soil dynamics, so this balances my first impression.

## MINOR COMMENTS

-----  
\* 1.53 and 1.80: "length  $\gg$  grain size d": It is true that, in other fields of condensed matter, a mesoscopic region will necessarily be much larger in size than the microscopic entity. But, for granular heaps, emulsions, as well as foams, the gap between length scales turns out to be much narrower; often, there is not even one order of magnitude difference between the size of the particle and that of the 'mesoscopic' zone of strain. Thus, I suggest simply writing "length  $>$  grain size d" or "length  $\sim$  several grain sizes d".

\* The manuscript often refers to Culling's theory of soil dynamics. Even if these references are mostly intended to call the theory into question, it is a pity not to have a slightly more detailed description of this theory, say, with one schematic equation. Would it be possible to include at least one schematic equation, or sketch, explaining the gist of the theory, e.g., in SI Section 1?

\* Figure S.9: Now, I understand that "C" stands for "Celsius". Still, I am not very accustomed to having axis labels with just the units, and not the variable name (nor to C instead of °C as an abbreviation, but this is less important).

\* Figure S.14 ("strain" correlations): The figure is beautiful, but (or perhaps "therefore") it deserves to be explained better: please add labels to the axes, specifying the length scales, and clarify what is plotted there exactly.

\* Finally, I am not quite convinced about the answer 3.E (6) to my question about the integration of the strain into the velocity (why illustrate a statement with this equation " $u = \epsilon l^*$ " if it is actually not used and it is not correct, which may raise doubts?).

Alexandre NICOLAS

schematic equations for Culling's theory

Reviewer #3 (Remarks to the Author):

I enjoyed reading the revised version of your manuscript and believe it to be an important contribution that would merit publication in Nature Communications.

My comments below mostly highlight discussion points that, in my view, would make your work more accessible to a broader audience.

I was left somewhat unsatisfied by your reply to my previous comment regarding the long-term effect of creep dynamics on hillslope evolution.

The abstract demonstrates that you see the importance in the long-term integrated effect: "...shaping landscapes and the human and ecological communities that live within them. What causes this granular material to 'flow'...".

Your contribution focuses on exceedingly slow hillslope dynamics. Suppose more rapid forms of hillslope motion are sufficiently frequent, i.e., frequent fluidized dynamics due to larger magnitude disturbances. In that case, this might overwhelm the creeping dynamics in terms of hillslope strain. (You show that Roering's fluidized regime also results in an exponential velocity-depth profile. Field strain data might not be able to differentiate between the two dynamics). I agree that the analysis is not trivial because of uncertainties and stochasticity. However, don't you agree that it is possible that in more disturbed landscapes, the contribution of creep dynamics is overwhelmed by more efficient transport modes? This could be a critical discussion point.

Your analogy to bedload flux exemplifies precisely this effect. If flux is dominated by grain motion during frequent large floods, then the contribution of a small occasional grain slip in between floods could be negligible.

I think that the effect of slope on creep should also go into this discussion point.

Line 59 – Wouldn't the geometry of 'flow down an inclined plane' (with  $\sim$ constant grain layer thickness) better represent natural hillslopes?

Line 73 – if I understand correctly, the creep motion you document is predominantly of the grain layer in contact with the acrylic wall. What could be the role of the wall in inducing creep? Grain-wall contacts are inherently different and possibly more susceptible to plastic relaxation than grain-grain contacts.

Lines 88-94 – Hard to follow without consulting the SI (where G is defined). A bit more intuition into the

meaning of  $G$ ,  $t$ , and  $\tau$  might make the reading smoother.

Same comment about Line 219. The equation was clarified only after reading the SI.

Liran Goren

# Responses to reviewer comments to Fragile Hillslopes

Nakul S. Deshpande1, David J. Furbish2,3, Paulo E. Arratia4, and Douglas J. Jerolmack1,4

1*Department of Earth and Environmental Science, University of Pennsylvania, Philadelphia, Pennsylvania, USA*

2*Departments of Earth and Environmental Sciences Vanderbilt University, Nashville, Tennessee, USA*

3*Civil and Environmental Engineering, Vanderbilt University, Nashville, Tennessee, USA*

4*Department of Mechanical Engineering & Applied Mechanics, University of Pennsylvania, Philadelphia, Pennsylvania, USA*

March 12, 2021

## 1 A case for revision and re-submission

We thank the editor for reconsidering our manuscript and returning it to the reviewers. We also thank the three reviewers for comments; these have greatly improved the quality and readability of our manuscript. We have addressed your further comments and edits below. Changes to the text are highlighted in red.

## 2 Reviewer 1 (Rachel Glade)

In my view, the authors have succeeded in addressing my initial relatively minor comments on this paper, and have put considerable work into addressing other reviewers' technical concerns. The paper is well written, represents a fundamental advance in both geomorphology and materials science, and will be of great interest to readers of Nature Communications. I have no revisions to suggest, and would be happy to see this published as is.

### 3 Reviewer 2 (Alexander Nicolas)

That the authors have taken great care to address the Reviewers' concerns in depth is greatly appreciated. (Besides, I am flattered that the authors mention that they have engaged with my previous work, but I hope that this sentiment does not interfere with my review.)

The (new) computation of the spatial correlations of their G function abates my major technical concern regarding the validity of their experimental setup. For sure, it does not fully annihilate this concern, because it is not entirely clear to me how accurate ("quantitative") the correspondence between G and the local strain is, but at least it successfully evinces a relation between the two variables.

Another issue that I had was in keeping with how compelling is the experimental demonstration of the 'mapping' between a soft matter problem (granular creep) and an ongoing geomorphology issue (soil creep). The previous version was very assertive in this regard. The more nuanced wording of the revised manuscript sounds much more acceptable to me and more reflective of the present findings. In addition, this toning down of the claims does not make the manuscript any less enjoyable to read.

Clearly, the manuscript brings interesting pieces of evidence in favour of the idea of such a 'mapping' between the two problems, but I side with Referee 3 in that it does not bring the debate to an end: Even if one accepts that external disturbances may rejuvenate the soil and thus maintain creep at non-vanishing level, one may wonder if the immediate effect of these external disturbances (thermal effects, freeze/thaw, tree throw, etc.) may not prevail over the subsequent, more persistent creep that they induce, as far as displacements are concerned.

Still, I agree that this vast debate can likely not be fully solved in a single paper and, therefore, I see no more objections to the publication of the manuscript in Nature Communications, as it is now (except for the minor revisions advocated below). My opinion is that researchers from the soft matter community might be more impressed by the sophisticated technical setup than surprised at the reported findings. However, the other reviewers have suggested that the echo may be much stronger (in terms of novelty) in the community investigating soil dynamics, so this balances my first impression.

#### 3.A Minor Comments

1. **1.53 and 1.80: "length  $\gg$  grain size  $d$ ":** It is true that, in other fields of condensed matter, a mesoscopic region will necessarily be much larger in size than the microscopic entity. But, for granular heaps, emulsions, as well as foams, the gap between length scales turns out to be much narrower; often, there is not even one order of magnitude difference between the size of the particle and that of the 'mesoscopic' zone of strain. Thus, I suggest simply writing "length  $>$  grain size  $d$ " or "length  $\sim$  several grain sizes  $d$ ".

Thank you for this clarification; we have indicated on Line 53 that:

length  $\sim$  grain size,  $d$

2. **The manuscript often refers to Culling's theory of soil dynamics. Even if these references are mostly intended to call the theory into question, it is a pity not to have a slightly more detailed description of this theory, say, with one schematic equation. Would it be possible to include at least one schematic equation, or sketch, explaining the gist of the theory, e.g., in SI Section 1?**

Thank you for your encouragement to include more of Culling's theory.

We have added the following section to the Supplementary Material S1:

Culling [1] had previously proposed that the granular flux  $q$  under the influence of gravity is described by a Fickian-like relation involving the land-surface slope,  $q = -K\partial\zeta/\partial x$ , which, when substituted into a statement of conservation leads to a diffusion equation,  $\partial\zeta/\partial t = K\partial^2\zeta/\partial x^2$ , of land-surface elevation  $\zeta$ . To justify this result, Culling [2] then followed the analysis of Einstein [3] and envisioned local Brownian-like particle displacements  $l$  involving the diffusivity  $D \sim n\langle l^2 \rangle$  with number displacement frequency  $n$ . He then appealed to a master equation describing changes in particle number density [4] and performed an approximate vertical integration over the soil column to obtain the desired result.

The formulation effectively envisions a slowly diffusing dense granular gas giving a bulk downslope motion due to the influence of gravity.

3. **Figure S.9:** Now, I understand that "C" stands for "Celsius". Still, I am not very accustomed to having axis labels with just the units, and not the variable name (nor to C instead of °C as an abbreviation, but this is less important).

Thank you for this clarifying point. We have changed the axis labels of Figure S.9 to state the quantities more clearly.

4. **Figure S.14 ("strain" correlations):** The figure is beautiful, but (or perhaps "therefore") it deserves to be explained better: please add labels to the axes, specifying the length scales, and clarify what is plotted there exactly.

Thank you for nothing the ambiguity in the figure. We have updated Figure S.14 and added the spatial coordinates. We have also added the functional form of the spatial correlation function to help explain what exactly is being shown.

5. **Finally, I am not quite convinced about the answer 3.E (6) to my question about the integration of the strain into the velocity (why illustrate a statement with this equation " $u = \epsilon l$ " if it is actually not used and it is not correct, which may raise doubts?).**

Our goal here is to roughly estimate the velocity for the purposes of comparison with field observations. You are absolutely correct: strain within a metapixel is not calculated relative to above and below – strain is measured within the metapixel relative to itself. Our strain rates are simply the total strain within a metapixel divided by the elapsed time; no integration of a velocity profile is performed. Our intent is to obtain the crudest order of magnitude estimate for velocity to ground the reader's intuition when comparing to field data (especially if the reader is a geoscientist). We also recognize going from strain rate to velocity requires assumptions that may not be valid. In our exercise we assume that all the motion within a metapixel is horizontal and that all decorrelation is all translational in the down-slope direction. Certainly, some strain must be due to differential motion, and some to translation.

## 4 Reviewer 3 (Liran Goren)

I enjoyed reading the revised version of your manuscript and believe it to be an important contribution that would merit publication in Nature Communications. My comments below mostly highlight discussion points that, in my view, would make your work more accessible to a broader audience. I was left somewhat unsatisfied by your reply to my previous comment regarding the long-term effect of creep dynamics on hillslope evolution. The abstract demonstrates that you see the importance in the long-term integrated effect: "...shaping landscapes and the human and ecological communities that live within them. What causes this granular material to 'flow'...".

### **Your contribution focuses on exceedingly slow hillslope dynamics.**

Indeed, our measurements are quite slow – but we would like to emphasize that true creeping motions in an undisturbed pile have not been reported experiments before our work. Just because they are slow does not mean they are not important: soil creep has been a touchstone concept in geomorphology since the time of Gilbert. The rates we measure here are comparable to many reported rates measured over longer timescales in natural landscapes.

### **Suppose more rapid forms of hillslope motion are sufficiently frequent, i.e., frequent fluidized dynamics due to larger magnitude disturbances.**

We fully acknowledge that our work does not deal with landslides, debris flows, or other fluidized phases of transport. We are wholly focused on creep. We note, however, that there are many hillslopes where landslides do not occur. Even in landscapes that are heavily scarred by landslides, significant area fractions of the landscape are hillslopes where landslides do not occur.

**In that case, this might overwhelm the creeping dynamics in terms of hillslope strain. (You show that Roering's fluidized regime also results in an exponential velocity-depth profile. Field strain data might not be able to differentiate between the two dynamics). I agree that the analysis is not trivial because of uncertainties and stochasticity. However, don't you agree that it is possible that in more disturbed landscapes, the contribution of creep dynamics is overwhelmed by more efficient transport modes? This could be a critical discussion point.**

First, we agree that exponential profiles are not diagnostic of transport mode. However, for Young pits measuring soil motion in the field, we are confident that this is creep and not fluidized motion — simply because the velocities are so slow that they are squarely in the creep regime. Indeed, this can be assessed by estimating the dimensionless shear rate  $I$ ; in Roering's experiments this was very large, which is why we said those flows were inertial. The values determined for soil Young pits in the field are very small, well below the value associated with transition to creep. For the sake of discussion we distinguish between two types of disturbances: those that create 'macroporosity' (tree throw, bioturbation, on the scale of the soil depth) and those which affect 'microporosity' (grain-scale rearrangements and disturbances). If disturbances on the scale of the soil depth dominate, then a continuum approach is not even relevant. Certainly, grain movement from those disturbances would not follow our creep (either in magnitude or in the shape of the velocity profile), but also no continuum model should be applicable (including any kind of linear or nonlinear 'diffusion'). But there are many hillslopes where soil-depth scale disturbances are not dominant. If microporosity is generated by any process - shrinking/swelling cycles, worms, etc. - this does not produce net downslope transport. Rather, downslope relaxation of the soil, to collapse/fill in that porosity, does. Even for larger porosity generators like animal burrows - unless the entire hillslope is covered in them and grains are quite literally being thrown downslope by animals, creep relaxation plays a fundamental role in downslope movement. In essence, they have a long-term consequence in that they are sources of rejuvenation. We have made this change to emphasize this point:

Line 145: **In such circumstances, macro-disturbances may locally and intermittently dominate downslope transport. Nonetheless, their role remains as a rejuvenating agent of the chronic creep and perpetual fragility.**

Your analogy to bedload flux exemplifies precisely this effect. If flux is dominated by grain motion during frequent large floods, then the contribution of a small occasional grain slip in between floods could be negligible. I think that the effect of slope on creep should also go into this discussion point.

Our analogy to bedload flux is identical to the transition from creep to landsliding. We agree that in landscapes where landslides are frequent, that they heavily contribute to soil flux and erosion. But in landscapes where landslides are rare or absent, creep dominates. This is the same separation that geomorphologists make when they model hillslope ‘diffusion’.

## 4.A Comments

1. **Line 59 – Wouldn’t the geometry of ‘flow down an inclined plane’ (with  $\sim$  constant grain layer thickness) better represent natural hillslopes?**

We note in the text (line 206) that our experimental setup is not intended to be a full replicate of a hillslope on the pedon scale. Basically, our setup represents an infinitely deep soil pile. One could instead choose an inclined plane – but this would be introducing a new lengthscale, the soil depth, which would also introduce some effect on dynamics. We are currently examining the controls of boundary conditions like soil depth, but it is not clear that choosing an arbitrary depth is a ‘better’ or more faithful choice. One could make an argument that not introducing an additional lengthscale to the problem is preferable, in the absence of some *a priori* knowledge.

2. **Line 73 – if I understand correctly, the creep motion you document is predominantly of the grain layer in contact with the acrylic wall. What could be the role of the wall in inducing creep? Grain-wall contacts are inherently different and possibly more susceptible to plastic relaxation than grain-grain contacts.**

Indeed, the role of walls and boundaries is important and necessary for creep. Walls and boundaries in the natural environment are manifest as bedrock contacts, mechanically varying soil horizons, vegetation roots, and so on. In our view there are three basic ingredients for creep to exist in the absence of external excitation: disorder, applied force (gravity) and walls. The walls are critical because they impose a boundary condition that prohibits the grains from fully exploring configurational space, and therefore the absolute energy minima. Yes, our DWS technique images grains close to the acrylic wall. Nonetheless, previous experimental and numerical work from our group ([5–7]) demonstrates that the grain dynamics in the bulk are similar to those near the wall; creep is not an artefact. To be clear, our measurements are in three dimensions and account for motions about three grain diameters away from the wall.

3. **Lines 88-94 – Hard to follow without consulting the SI (where  $G$  is defined). A bit more intuition into the meaning of  $G$ ,  $t$ , and  $\tau$  might make the reading smoother.**

We have added the additional sentence to try and bridge the intuition of the reader and the detail given in the methods (Line 89):

*$\langle G \rangle$  is in essence an auto-correlation function which charts how a configuration of grains (speckles) changes in time ( $\tau$ ) with respect to a configuration at start time  $t$ .*

4. **Same comment about Line 219. The equation was clarified only after reading the SI.**

In the text we are avoiding being overly technical.

## References

1. Culling, W. E. H. Analytical Theory of Erosion. *The Journal of Geology* **68**, 336–344 (1960).
2. Culling, W. E. H. Soil Creep and the Development of Hillside Slopes. *The Journal of Geology* **71**, 127–161 (1963).

3. Einstein, A. On the Motion Required by the Molecular Kinetic Theory of Heat of Small Particles Suspended in a Stationary Liquid. *Annalen der physik* **17**, 549–560 (1905).
4. Chandrasekhar, S. Stochastic Problems in Physics and Astronomy. *Reviews of modern physics* **15**, 1 (1943).
5. Houssais, M., Ortiz, C. P., Durian, D. J. & Jerolmack, D. J. Onset of Sediment Transport Is a Continuous Transition Driven by Fluid Shear and Granular Creep. *Nature Communications* **6**, 1–8 (2015).
6. Houssais, M., Ortiz, C. P., Durian, D. J. & Jerolmack, D. J. Rheology of Sediment Transported by a Laminar Flow. *Physical Review E* **94**, 062609 (2016).
7. Ferdowsi, B., Ortiz, C. P. & Jerolmack, D. J. Glassy Dynamics of Landscape Evolution. *Proceedings of the National Academy of Sciences* **115**, 4827–4832 (2018).